# Contrast polarity-specific mapping improves efficiency of neuronal computation for collision detection

**Richard Burkett Dewell[1]\*[†], Ying Zhu[1][†][‡], Margaret Eisenbrandt[1], Richard Morse[2], Fabrizio Gabbiani[1]**

[1]Department of Neuroscience, Baylor College of Medicine, Houston, United States; [2]Rice University, Houston, United States

**Abstract** Neurons receive information through their synaptic inputs, but the functional significance of how those inputs are mapped on to a cell's dendrites remains unclear. We studied this question in a grasshopper visual neuron that tracks approaching objects and triggers escape behavior before an impending collision. In response to black approaching objects, the neuron receives OFF excitatory inputs that form a retinotopic map of the visual field onto compartmentalized, distal dendrites. Subsequent processing of these OFF inputs by active membrane conductances allows the neuron to discriminate the spatial coherence of such stimuli. In contrast, we show that ON excitatory synaptic inputs activated by white approaching objects map in a random manner onto a more proximal dendritic field of the same neuron. The lack of retinotopic synaptic arrangement results in the neuron's inability to discriminate the coherence of white approaching stimuli. Yet, the neuron retains the ability to discriminate stimulus coherence for checkered stimuli of mixed ON/OFF polarity. The coarser mapping and processing of ON stimuli thus has a minimal impact, while reducing the total energetic cost of the circuit. Further, we show that these differences in ON/OFF neuronal processing are behaviorally relevant, being tightly correlated with the animal's escape behavior to light and dark stimuli of variable coherence. Our results show that the synaptic mapping of excitatory inputs affects the fine stimulus discrimination ability of single neurons and document the resulting functional impact on behavior.

**\*For correspondence:**
dewell@bcm.edu

[†]These authors contributed equally to this work

**Present address:** [‡]MD Anderson Cancer Center, Houston, United States

**Competing interest:** The authors declare that no competing interests exist.

## Editor's evaluation

This valuable article will be of interest to neuroscientists who study visual processing or are interested in dendritic integration. The authors used calcium imaging, pharmacology, and electrophysiology to investigate how a large, loom-sensitive neuron in grasshoppers integrates visual input to respond to both light and dark looming objects. These experiments convincingly support the finding that the integration is done by two distinct arbors of the neuronal dendritic tree, one of which loses retinotopic information. The authors suggest energetic advantages of this dendritic arrangement.

## Introduction

A major goal of neuroscience is determining the mechanisms of neural computation – how is information processed within neural circuits and how cellular properties produce these abilities. Neurons receive information about the external world through the pattern of their synaptic inputs. So, determining how they integrate them is critical. For some tasks, such as assessing the presence of an impending threat, sensory discriminations must be quick and reliable for survival. In neural processing, there are trade-offs, though, between speed, high-resolution discrimination, and energy efficiency

(*Attwell and Laughlin, 2001*; *Laughlin, 2001*; *Vincent and Baddeley, 2003*; *Hasenstaub et al., 2010*). Evolution presumably settles on solutions that trend towards minimizing energy expenditure while maintaining required processing abilities.

Individual neurons process information by discriminating between patterns of synaptic inputs impinging on their dendrites. Presynaptic targeting, dendritic morphology, compartmentalization, and active membrane properties all shape neuronal processing (*London and Häusser, 2005*; *Major et al., 2013*; *Lefebvre et al., 2015*; *Hawkins and Ahmad, 2016*). Many neurons, including the principal excitatory neurons of the mammalian cortex, pyramidal neurons, have distinct dendritic subfields proximal and distal to the site of spike initiation that receive functionally segregated synaptic inputs processed by distinct membrane conductances (*Behabadi et al., 2012*; *Major et al., 2013*; *Hawkins and Ahmad, 2016*). In these neurons, the role of anatomical segregation within dendrites in integrating inputs from distinct neural pathways remains unclear.

In the visual system, neural channels transmitting information about luminance increases (ON) and decreases (OFF) constitute one key example of parallel pathways requiring integration (*Chen et al., 2017*; *Williams et al., 2021*). Although luminance information is largely segregated between ON and OFF cells early on in the retina of mammals and in insect visual neuropils, real-world scenes involve a mix of both (*Leonhardt et al., 2016*; *Chen et al., 2019*; *Mazade et al., 2019*; *Mulholland and Smith, 2021*). The visual world contains more information in luminance decreases, particularly for smaller, faster objects (*Simoncelli and Olshausen, 2001*; *Clark et al., 2014*; *Leonhardt et al., 2016*; *Chen et al., 2019*). Although the contrast statistics of approaching predators has not been determined, it is likely that they also contain more OFF than ON information (*Zhou et al., 2022*). How these pathways are integrated within individual neurons driving behavior remains unanswered, though.

To address these questions, we leveraged a well-studied, identified neuron in grasshoppers (*Schistocerca americana*) that receives ON and OFF inputs across three distinct dendritic subfields. It processes these inputs to detect impending collision, resulting in a spiking output critical to the generation of escape behavior (*O'Shea and Williams, 1974*; *Gabbiani et al., 1999*; *Fotowat et al., 2011*; *Dewell and Gabbiani, 2018a*). This so-called lobula giant movement detector (LGMD) neuron receives visual information from all ommatidia (facets) of the ipsilateral compound eye after processing in two intermediate neuropils. It is about as large as a human cerebellar Purkinje cell and its output is faithfully relayed by a second neuron, the descending contralateral movement detector (DCMD), to the cells coordinating escape jumps (*O'Shea and Williams, 1974*).

The LGMD receives retinotopic medullary excitation from columnar projections to dendritic field A through the second, crossed optic chiasm (*Strausfeld and Nässel, 1981*; *Peron et al., 2009*; *Zhu and Gabbiani, 2016*). Dendritic fields B and C receive non-retinotopic ON and OFF inhibition, respectively (*Fraser Rowell et al., 1977*; *Strausfeld and Nässel, 1981*). In particular, dendritic field C arborizes in a distinct sub-compartment of the lobula, called the dorsal lobula (*O'Shea and Williams, 1974*; *Rosner et al., 2017*). The projections to field C arise from the dorsal uncrossed bundle (DUB), which contains a subpopulation of ~500 neurons with ~10° receptive fields impinging non-retinotopically to dendritic field C (*Gouranton, 1964*; *Strausfeld and Nässel, 1981*; *Elphick et al., 1996*). More recent work has shown that the OFF inhibitory inputs to field C have wider receptive fields (~50°) and that only a small subset of the DUB axons likely encodes this inhibitory signal, raising the possibility that the DUB contains more than one type of projection to field C (*Wang et al., 2018b*).

The LGMD distinguishes the coherence of dark (OFF) approaching objects through a retinotopic mapping of synaptic inputs onto compartmentalized field A dendrites and through intracellular processing by differentially distributed active conductances (*Dewell and Gabbiani, 2018a*; *Zhu et al., 2018*). These include an H-conductance and an inactivating $K^+$ conductance localized in dendrites, as well as calcium and calcium-sensitive $K^+$ conductances localized close to the spike initiation zone (SIZ), and an M-type $K^+$ conductance presumably localized in the axon (*Peron and Gabbiani, 2009*; *Dewell and Gabbiani, 2018a*; *Dewell and Gabbiani, 2018b*). Little is known about the LGMD's processing of ON inputs.

Using calcium imaging, electrophysiology, pharmacology, and modeling, we characterized a novel excitatory pathway to the LGMD for ON inputs. We show that the mapping of ON and OFF excitation to distinct dendritic fields has behavioral consequences. Further, we show that the mechanisms of ON–OFF integration within the LGMD's dendritic arbor allows the detection of approaching objects of any contrast polarity, while maintaining in most cases selectivity for their spatial coherence. Finally,

we show that the mapping of ON and OFF excitatory inputs on distinct dendritic subfields allows the implementation of coherence discrimination in an energy-efficient manner, suggesting integration principles likely applicable to other neurons, including within our own brains.

## Results

Visual circuits are subdivided into ON and OFF pathways, but animals can face approaching threats that are light, dark, or a mottled combination (*Strother et al., 2014*; *Dunn et al., 2016*; *Chen et al., 2019*; *Branco and Redgrave, 2020*). Although extensive work has investigated the role of OFF pathways in detecting impending collision, how ON and OFF signals are integrated to decide whether to initiate escape is unknown. We used grasshoppers that have well-characterized jump escape behaviors initiated by an identified neuron in the optic lobe to explore the neuronal integration of ON and OFF contrast polarities for collision avoidance.

### Influence of contrast polarity on jump probability and timing

In grasshoppers and other species, there has been extensive research on the behavioral responses produced by black looming stimuli (*Fotowat et al., 2011*; *Yilmaz and Meister, 2013*; *Dunn et al., 2016*; *Dewell and Gabbiani, 2018a*; *Heap et al., 2018*). In comparison, there has been little investigation into how behavior depends on contrast polarity although mice and house flies escape more often from simulated black approaching stimuli (*Holmqvist and Srinivasan, 1991*; *Yilmaz and Meister, 2013*). First, we tested whether the animals escape from simulated white objects approaching at constant speeds as they do for black ones (*Figure 1A*, *Video 1*). These looming stimuli are characterized by the parameter $l/|v|$ (i.e., the ratio of their half-size to approach speed; see *Figure 1A*; *Gabbiani et al., 1999*).

Grasshoppers jumped in response to all looming stimuli, escaping from 57% of black (OFF) looms and 45% of white (ON) looms (p=0.065; *Figure 1B*). For neither contrast polarity did the approach speed influence jump probability (for a fixed stimulus size; p=0.77 for white, p=0.42 for black, Kruskal–Wallis test [KW]). Jump probabilities were higher for OFF looms in six of the seven animals tested (p=0.047; *Figure 1C*). The jump timing changed slightly with contrast polarity, however, with white stimuli producing earlier jumps for intermediate approach speeds (*Figure 1D*). Specifically, for looming stimuli with an $l/|v|$ of 80 ms the median jump time was 223 ms earlier (p=8.9 • 10$^{-4}$); the jump time was also earlier, but not significantly so for white looms with an $l/|v|$ of 40 or 120 ms. All stimulus times were recorded relative to the projected time of collision, when the stimulus would reach a full angle of 180°.

### Dendritic field segregation of ON/OFF excitatory inputs to the LGMD neuron

In grasshoppers, previous investigations into the visual detection of impending collisions have focused on black looming stimuli, generating a detailed characterization of their processing by the presynaptic circuitry to the LGMD and by active membrane conductances within the LGMD (*Peron and Gabbiani, 2009*; *Jones and Gabbiani, 2010*; *Dewell and Gabbiani, 2018a*). The LGMD has a large distal dendritic field (field A) and two smaller ones more proximal to the SIZ (fields B and C; *Figure 2A*). Inhibitory inputs to the LGMD arise from pathways segregated by contrast polarity, with OFF inhibition impinging on dendritic field C and ON inhibition impinging on field B (*Fraser Rowell et al., 1977*; *Strausfeld and Nässel, 1981*). The field C inhibitory inputs were recently found to be conveyed by widefield neurons likely comprising only a small fraction of the impinging DUB axons, raising the possibility that it might receive additional uncharacterized inputs (*Wang et al., 2018a*).

The LGMD's excitatory inputs, however, were believed to be processed independent of polarity, with both ON and OFF pathways projecting retinotopically to field A (*Figure 1Ai*, *Figure 2A*, colored dots; *O'Shea and Rowell, 1976*). This arrangement has been documented for OFF, but not for ON stimuli (*Peron et al., 2009*; *Zhu and Gabbiani, 2016*). We tested its validity for ON inputs with widefield calcium imaging of all three dendritic fields during presentation of light and dark looming stimuli (*Figure 2*).

The previously described excitatory synaptic inputs onto field A are mediated through nicotinic acetylcholine receptor (nAChR) channels that are calcium permeable (*Peron et al., 2009*). As the dendrites lack voltage-gated $Ca^{2+}$ channels, this allows direct imaging of synaptic activation with an

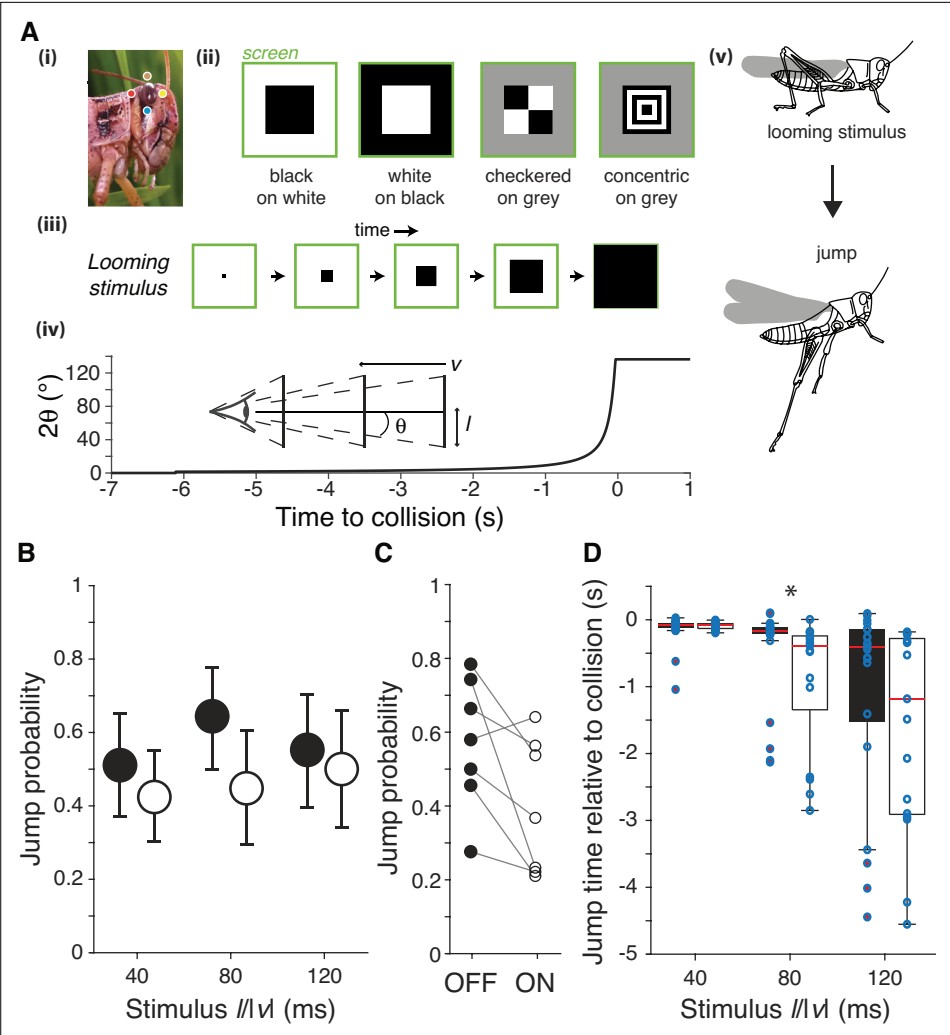

**Figure 1.** Escape behavior to white and black looming stimuli. (**A**) The locust eye (i) was stimulated using four simulated approaching squares (looming stimuli) with different contrast polarities (ii). (iii, iv): schematic of looming stimulus expansion on the retina (in the case of a black square) and time course of angular expansion for $l/|v| =$ 80 ms, respectively. Inset illustrates the definition of the approach speed ($v < 0$), the square half-size (l) and the half-angle of the stimulus subtended at the retina ($\theta$). Behaviorally, such stimuli lead to escape jumping (v). In (i), colored dots match those of *Figure 2A*. (**B**) Grasshoppers consistently jumped to white and black stimuli of different $l/|v|$. White stimuli had slightly lower but not significantly reduced escape responses (p=0.065, Fisher's exact test). Error bars are 95% confidence intervals. (**C**) Comparison of jump response polarity preference for the seven animals tested shows reduced escape jumps for white stimuli (p=0.047, Wilcoxon signed-rank). (**D**) White stimuli produced jumps earlier relative to collision than did black stimuli for $l/|v|$ of 80 ms; for $l/|v|$ of 40, 80, and 120 ms, p=0.83, 8.9 • 10⁻⁴, and 0.26, respectively (Wilcoxon rank-sum test [WRS]). Points show individual trials. For (**B**) and (**C**), N = 263 trials from seven grasshoppers. See Methods, data analysis and statistics, for a description of box plot conventions.

intracellularly injected fluorescent calcium dye, while voltage-gated $Ca^{2+}$ channels near the SIZ allow imaging of the neuron's SIZ activation (*Peron and Gabbiani, 2009*; *Peron et al., 2009*). Presentation of an OFF-looming stimulus produced the expected increase in fluorescence in field A (*Figure 2B*). In response to an ON-looming stimulus, however, fluorescence increased in field C more than in field A (*Figure 2C*, *Video 2*). No fluorescence changes were observed in field B, regardless of stimulus contrast polarity. The time course of the fluorescence increase was similar in fields A and C and near the SIZ with a clear difference in dF/F amplitude depending on contrast polarity (*Figure 2D, E*, *Videos 3 and 4*). This provides the first evidence that excitation from ON and OFF inputs is largely segregated and independently integrated within the LGMD.

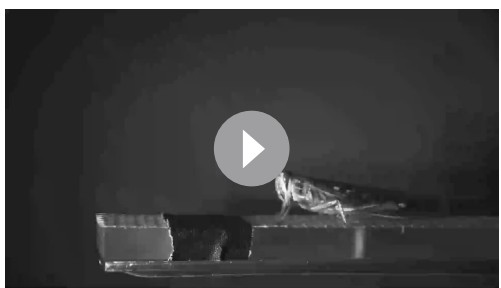

**Video 1.** Example jump escape response of a grasshopper to a white looming stimulus. As the stimulus expands, the animal jumps and flies away before the time of projected collision. Frame rate: 200 Hz; slowed to 30 Hz. Compare to a similar response for a black looming stimulus in Video 3 of *Dewell and Gabbiani, 2018a*; DOI: 10.7554/eLife.34238.008.
https://elifesciences.org/articles/79772/figures#video1

For comparison between animals, we normalized each animal's fluorescence signal to the maximum response to an ON loom in field C. During OFF-looming expansion, field A consistently had a larger response, while field C responded more to ON looms (*Figure 2F*). Following looming stimuli, the screen changed abruptly back to its original background luminance after ~2 s causing a 'flash' of the opposite polarity to the loom. ON flashes produced a larger response in field C and OFF flashes a larger response in field A (*Figure 2F and G*). For looms of either polarity, the peak fluorescence change occurred near the projected time of collision, with the peak response in field A trailing the peak fluorescence in field C and the SIZ (*Figure 2H*). The average peak times were very similar in each ROI for ON and OFF looms (*Figure 2—figure supplement 1*).

To characterize the consistency in polarity segregation across animals, we compared peak responses between dendritic fields for ON and OFF stimuli (*Figure 2I, J*) and the ratio of peak responses between ON and OFF stimuli within each dendritic field (*Figure 2K, L*). ON looms produced field C responses 2.4 ± 1.2 times that of OFF looms, while in field A OFF loom responses were 3.6 ± 1.1 times those of ON looms (mean ± SD). Flash responses showed a stronger preference with responses 3.4 ± 1.7 times higher for ON than for OFF flashes in field C and OFF flash responses 5.6 ± 2.2 times that of ON flashes in field A. Calcium fluorescence at the SIZ did not consistently differ with stimulus polarity for either loom or flash stimuli (*Figure 2I–L*).

## ON field C synaptic inputs are mediated by nicotinic acetylcholine receptors

Having discovered an unexpected ON pathway projection onto field C of the LGMD, we tested how similar the ON inputs to field C are to the previously characterized OFF inputs in field A. Since the field A inputs are mediated by nAChRs, we examined whether that was also the case for the ON inputs to field C. Direct iontophoresis of acetylcholine (ACh) to field C dendrites increased calcium fluorescence at the location of ACh application dependent on the amount of applied ACh (*Figure 3A, B*). Across experiments, the LGMD $Ca^{2+}$ influx in field C following iontophoresis was consistently limited to the site of direct ACh exposure showing that the ACh activation was most likely directly through receptors within field C (*Figure 3—figure supplement 1*). That these field C receptors are nicotinic was confirmed with the use of the nAChR antagonist mecamylamine. Local puffing of mecamylamine near field C reduced calcium fluorescence produced by looming stimuli within field C and at the SIZ (*Figure 3C–F*). Mecamylamine reduced peak dF/F in field C by 80 ± 8.5% for ON looms and 75 ± 17% for OFF looms (mean ± SD).

## ON field C synaptic inputs are not retinotopically distributed

Excitatory inputs impinging on field A follow a precise retinotopic arrangement (*Peron et al., 2009*; *Zhu et al., 2018*). To test whether this was also the case for field C excitation, we examined the timing of activation within dendritic subregions (*Figure 4A*). Both ON and OFF looming stimuli produced synchronous activation of field C dendrites (*Figure 4B, C*). This differs from field A, which exhibits sequential activation of dendritic regions as the loom expands (*Figure 4D*; *Zhu et al., 2018*). This difference between the two dendritic fields was even more pronounced for the responses to translating squares (*Videos 5 and 6*).

Field A showed sequential activation of subregions as the visual object translated across the LGMD receptive field (*Figure 4E*, *Video 5*). Within field C, however, all dendritic subregions were activated simultaneously (*Figure 4F*, *Videos 6 and 7*). This was quantified by measuring the range of peak times across subregions (*Figure 4*). Despite field C being subdivided into more dendritic subregions

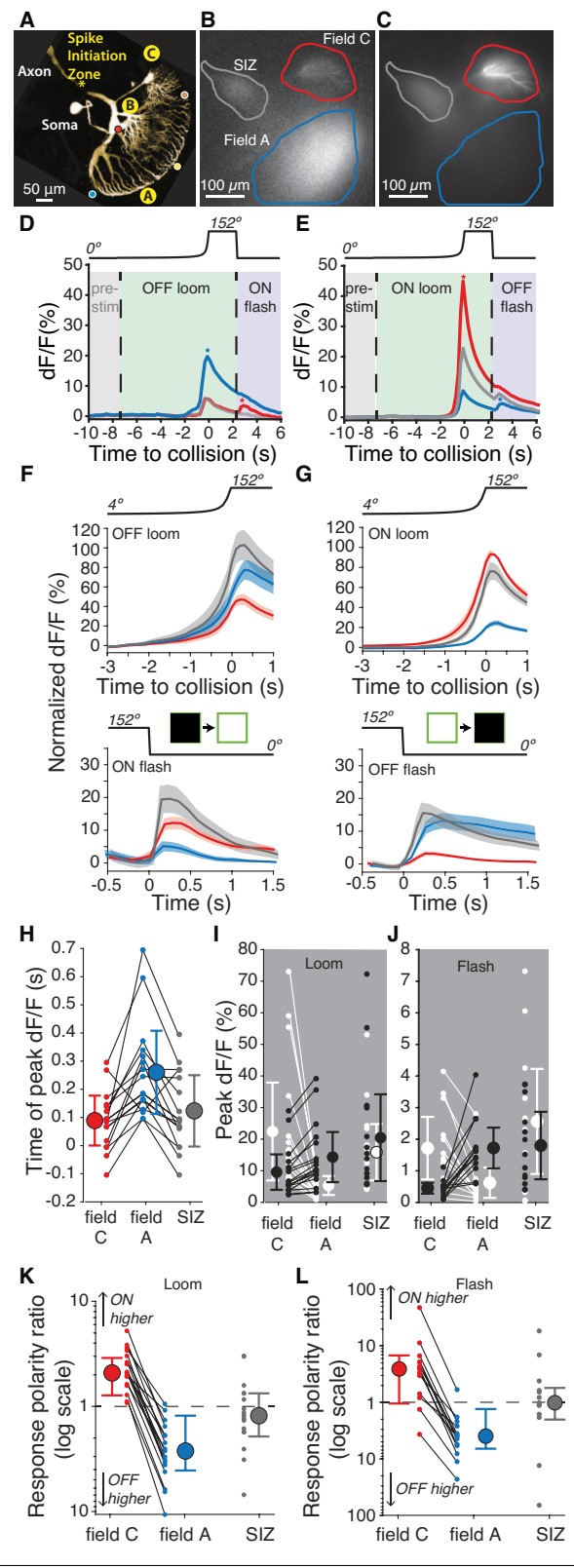

**Figure 2.** Luminance increases produce excitation in dendritic field C of the lobula giant movement detector (LGMD), unlike field A. (**A**) Micrograph (two-photon scan) of the LGMD neuron illustrating the neurites imaged: dendritic fields A and C, as well as the spike initiation zone (SIZ). Colored dots matching those of *Figure 1Ai* indicate retinotopic mapping of visual field onto LGMD dendritic field A. (**B, C**) Micrographs of

*Figure 2 continued on next page*

*Figure 2 continued*

maximum dF/F taken during presentation of OFF and ON, looming stimuli. Areas enclosed by solid lines are regions of interest (ROIs) used to compute dF/F time course. (**D, E**) Corresponding dF/F time course in response to an OFF, respectively ON, looming stimulus within the ROIs marked by matching colored lines in (**B, C**). Pre-stim, baseline prior to looming; ON flash, disappearance of full-size black square from screen. Field A showed a larger increase in dF/F during looming (blue star, **D**). Field C showed a larger rebound to the ON flash (red star, **D**). Conversely, the ON loom produced a larger dF/F in field C (red star, **E**). Its disappearance (OFF flash) produced a larger rebound dF/F in field A (blue star, **E**). (**F**) Population average normalized dF/F for the three ROIs during presentation of black stimuli (same color code as in **B**). Lines and shaded regions are mean ± SEM. In each animal, normalization is to the peak response for ON looms. (**G**) Similar average normalized dF/F for ON stimuli. (**H**) The peak dF/F in field A occurred later than in field C for both ON and OFF looms (p=2.7 • $10^{-4}$). (**I**) Peak dF/F response to ON and OFF looming stimuli for each animal. Responses to ON stimuli were higher in field C (p=1.5 • $10^{-5}$) and black looming responses were higher in field A (p=1.5 • $10^{-5}$). (**J**) Peak dF/F response to post-loom flash. Responses to ON flashes were higher in field C (p=2.4 • $10^{-4}$) while OFF flash responses were higher in field A (p=2.4 • $10^{-4}$). (**K**) For all animals, loom response polarity is biased towards ON stimuli in field C (p=1.5 • $10^{-5}$). In field A, it is biased towards OFF stimuli in 16 of 17 animals (p=2.7 • $10^{-4}$). (**L**) Similarly, response polarity is biased towards ON flashes in field C (p=2.3 • $10^{-3}$) and towards OFF flashes in field A (p=2.7 • $10^{-4}$). For (**F–L**), N=17, and colors in (**D–H**) and (**K, L**) match the ROIs shown in (**B**). SIZ is gray, field A is blue, and field C is red. The p-values for (**H–L**) are from two-sided sign tests.

The online version of this article includes the following figure supplement(s) for figure 2:

**Figure supplement 1.** Timing of peak dF/F within each dendritic field as shown in *Figure 2H* with responses separated for ON (**A**) and OFF (**B**) looms.

than field A, thus biasing it toward a larger range, it consistently had more synchronous activation of dendritic regions than field A. For 12 of 18 animals, all field C subregions had the same dF/F peak time in response to ON looms, and in no experiment did the responses suggest a retinotopic mapping of inputs to field C. Thus, the synaptic mapping of ON inputs onto field C is strikingly different from the previously described, ordered retinotopic organization of OFF synaptic inputs onto field A.

To further test whether the field C mapping was random, we computed the correlation between the response center of mass (CoM) and stimulus position for small translating stimuli. For field A, the CoM of dF/F moved across the dendritic arbor as the stimulus traversed the visual field (*Figure 4*, blue lines). For all trials tested, the CoM position was correlated to stimulus position (p<0.001). For field C, 9 of 21 trials had significant correlations between stimulus position and response CoM, but the CoM trajectories were short, winding, and inconsistent across trials within the same animal (*Figure 4H*, red lines). A 80° change in stimulus position produced a mean shift in CoM of 3.5 ± 5.6 µm (mean ± SD) in one dimension and 1.6

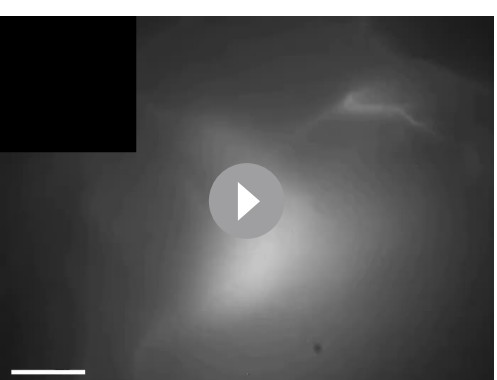

**Video 2.** Example of raw fluorescence signal within the lobula giant movement detector (LGMD) in response to a white looming stimulus (inset upper left). As the looming stimulus expands, note the fluorescence increase in field C (top right, in focus). Baseline field A fluorescence is also visible at the center. Frame rate for *Videos 2–7*: 5 Hz; real speed. The visual stimuli in *Videos 2–7* have been downsampled from 240 to 5 Hz to match fluorescence data. Matches data of *Figure 2C and E*. Image size for *Videos 2–7*: 630 × 470 µm. Scale bar: 100 µm.
https://elifesciences.org/articles/79772/figures#video2

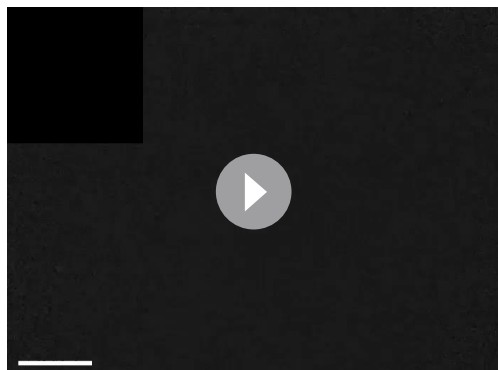

**Video 3.** The calculated fluorescence dF/F of the same trial shown in *Video 2*. Note the disappearance of field A baseline fluorescence through this manipulation.
https://elifesciences.org/articles/79772/figures#video3

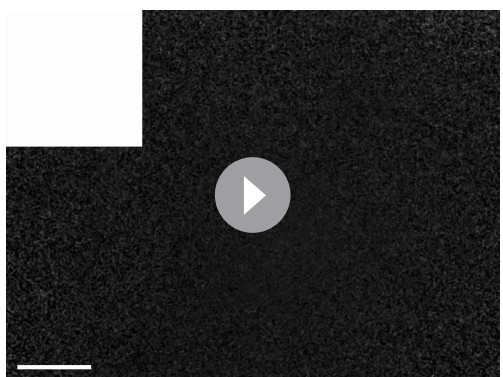

**Video 4.** An example of the fluorescence dF/F during a black looming stimulus (inset upper left). Data is from the same lobula giant movement detector (LGMD) as shown in *Videos 2 and 3*. Matches data of *Figure 2B, D*. Scale bar: 100 µm.

https://elifesciences.org/articles/79772/figures#video4

± 5.0 µm in the second dimension within field C (*Figure 4H*, right). In comparison, the same change in stimulus position resulted in a mean shift in CoM of 64.5 ± 25.8 µm and 84.3 ± 31 µm for the respective axes in field A. In addition to the smaller magnitude of CoM change, the direction of change was not consistent for either dimension within field C ($p_{ST}$ = 0.65 and 0.064 for the respective dimensions) unlike for field A ($p_{ST}$ = 0.0005 for each dimension).

We also compared the CoM trajectories to those of a pixelwise randomization of the responses (*Figure 4H*, green and yellow lines). As expected, CoM trajectories in field A correlated very strongly with stimulus positions (mean correlation coefficients of 0.98 and –0.95) and were significantly different from random (*Figure 4I*). Field C had weaker correlations (mean coefficients of –0.01 and 0.48) and along one of two axes the CoM trajectories were not different from a randomly shuffled version ($p_{RS}$ = 0.68 and 0.006). The nonrandom change was observed in the downward direction and was small (see *Figure 4H*, red lines). The animal with the strongest correlation between field C response CoM and stimulus position is shown in *Video 6*, where it is clear that the correlation is due to a spread in activation rather than activation of distinct regions. In summary, the mapping in field C appears nearly uniform across dendrites, except for slight variations at length scales smaller than 3.5 µm. This indicates that there is no retinotopic mapping of field C excitation.

## Absence of ON spatial coherence sensitivity in LGMD and behavior

What are the consequences of the non-retinotopic mapping of ON synaptic inputs to the LGMD? Earlier work showed that in field A both the retinotopic mapping of excitatory inputs and active dendritic processing are critical for grasshoppers' ability to discriminate the spatial coherence of black looming stimuli, a computation akin to object segmentation (*Dewell and Gabbiani, 2018a*). The large HCN conductance in field A that is necessary for discriminating spatial coherence of black looms is absent in field C, but it is otherwise unknown whether field C harbors active conductances (*Dewell and Gabbiani, 2018a*). We used electrophysiology to test whether the lack of retinotopy and HCN conductances in field C prevented discrimination of the spatial coherence of white looms. For this purpose, we used the same approach as previously used for black stimuli (*Dewell and Gabbiani, 2018a*). We first pixelated the screen at the spatial resolution of photoreceptor receptive fields and replaced local edge motion by an equivalent local luminance change in each pixel to obtain 'coarse looming stimuli' (*Figure 5A*, middle). We then randomly displaced each pixel at increasingly distant locations to obtain stimuli of decreasing spatial coherence (*Figure 5A*, bottom). Indeed, and in contrast to black looms, the peak firing rate of the LGMD did not change with the spatial coherence of white looms. However, the responses during the last second before collision contained more bursts for incoherent looms, a change in firing previously found to decrease escape responses for black looms (*Figure 5B*; *Dewell and Gabbiani, 2018a*).

This difference in stimulus coherence sensitivity for white and black looms raised the question of how the LGMD would respond to stimuli containing a mixture of ON and OFF polarities, such as checkerboard stimuli (*Figure 1A*). For spatially coherent stimuli, the peak firing rate for checkered stimuli was higher and occurred later than for solid white stimuli (p=0.001, sign-rank test). Reducing the coherence of black and white checkered looming stimuli decreased LGMD firing (*Figure 5C*), as observed for solid black looms (*Figure 5—figure supplement 1*). Of the nine animals studied, only three responded maximally to the 100% coherent stimulus for white looms (*Figure 5D*), while all nine animals responded maximally to the 100% coherent stimulus for checkered looms (*Figure 5E*). Thus, the spatial coherence preference for checkered looms was similar to that of black looms characterized

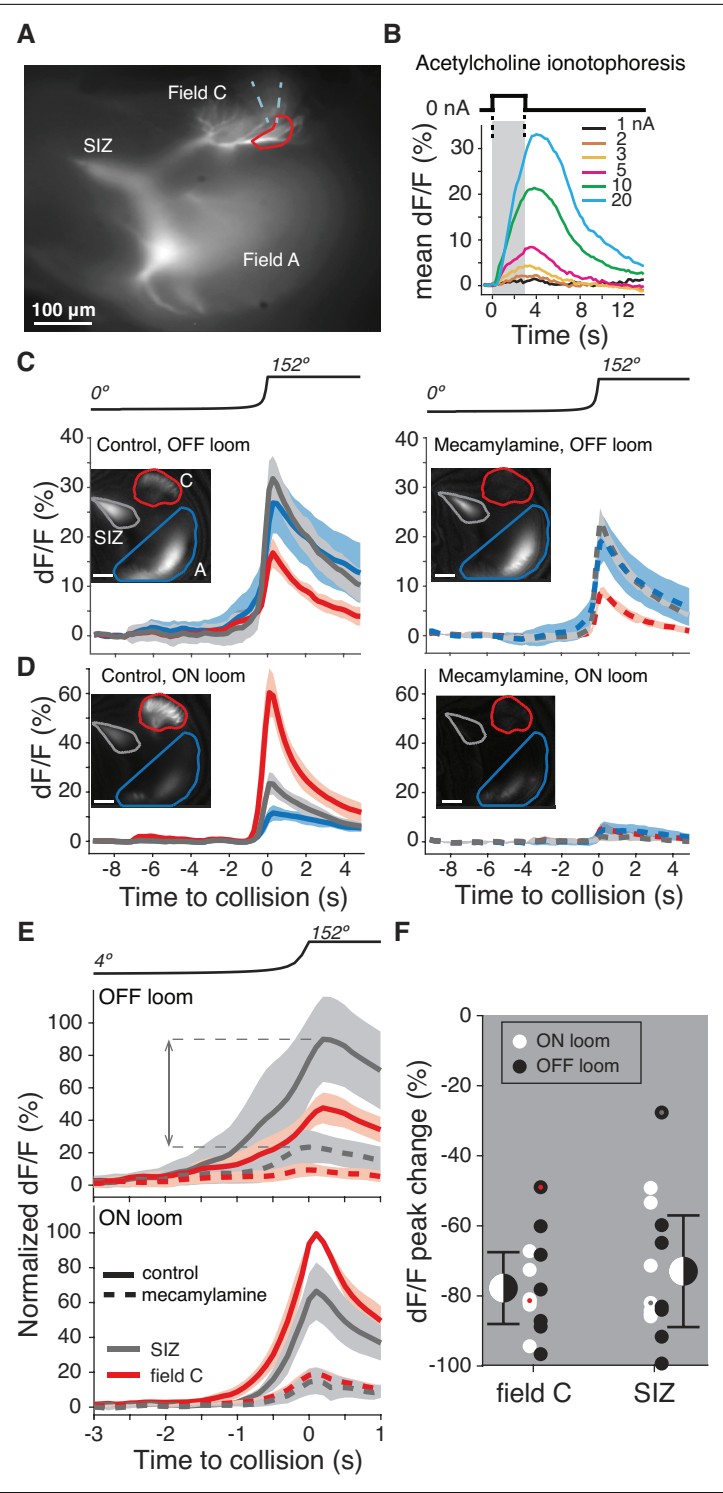

**Figure 3.** Dendritic field C receives excitatory synaptic inputs through nicotinic acetylcholine receptors (nAChRs). (**A**) Micrograph of the lobula giant movement detector (LGMD) stained with OGB1 showing the locations of the spike initiation zone (SIZ), dendritic field A (both out of focus), and dendritic field C (in focus). Dashed blue wedge: location of iontophoresis electrode; closed red curve: boundary of area used to compute dF/F in (**B**). (**B**) Example dF/F to ACh iontophoresis for square current pulses (1–20 nA; top inset and gray shading). (**C, D**) Left: example responses from one animal showing the dF/F produced by $Ca^{2+}$ influx in response to an OFF (**C**) or ON (**D**) looming stimulus in all three LGMD subregions (insets, as in **A**). Right: application of the nAChR blocker mecamylamine to field C reduced this influx (pipettes were placed as shown in **A**). Lines and shaded region are trial mean ± SD.

*Figure 3 continued on next page*

*Figure 3 continued*

(**E**) Average dF/F (± SEM) of seven animals before (solid lines) and after mecamylamine application (dashed lines) in field C (red) and at the SIZ (gray). Responses were normalized to each animal's mean peak dF/F response in field C during an ON looming stimulus. Dashed horizontal lines and double arrow show how the peak change plotted in (**F**) was computed. (**F**) Looming responses were reduced in field C by 78% (p=0.015, sign test [ST]) and by 73% at the SIZ (p=0.015, ST) on average. Field A dF/F was reduced as well, 58%, but less than field C (p=0.013, ST). Black and white dots: individual animal responses. Half black and white discs: mean ± SD pooled across animals and stimuli. Dots marked red (field C) and gray (SIZ) correspond to the animal shown in (**C, D**).

The online version of this article includes the following figure supplement(s) for figure 3:

**Figure supplement 1.** Micrographs of OGB1 fluorescence during iontophoresis of acetylcholine (ACh).

by *Dewell and Gabbiani, 2018a* (*Figure 5E*, right inset). The peak calcium fluorescence in field C was also unchanged by the spatial coherence of ON looms (*Figure 5F*).

In response to black looms, the jump escape probability of grasshoppers is exquisitely sensitive to stimulus coherence and tightly correlated with the peak firing rate response of the LGMD (*Dewell and Gabbiani, 2018a*). To determine whether the grasshoppers' escape behavior depended on the spatial coherence of white stimuli, we presented the same stimuli to freely moving animals. The jump probability showed a slight decrease with reduced spatial coherence, but nothing like the steep coherence preference seen previously for black looming stimuli (*Figure 5G*, *Video 1* and *Video 8*). Thus, this reduced ability to discriminate the spatial coherence of an approaching white stimulus is presumably due to the excitatory ON inputs impinging non-retinotopically onto field C and a lack of HCN conductances devoted to processing them there (*Figure 4*).

## Similar LGMD calcium responses for black and various mixed ON/OFF stimuli

So far, we have seen that ON stimuli excite dendritic field C non-retinotopically (*Figure 2*), which leads to a lack of selectivity for the spatial coherence of ON looms (*Figure 5*). But many real-world predators would contain a mix of light and dark regions, which would excite both dendritic fields. To measure how stimuli containing both ON and OFF inputs are processed, we imaged the calcium influx produced by approaching dark and light checkerboards and concentric squares on 50% luminance backgrounds (*Figure 6A, B*). The dF/F was higher in field A than in field C during approach for both stimuli and the post-loom flash response was higher in field C (*Figure 6C and D*), similar to the responses for solid black squares on a white background (*Figure 2F*). For the seven animals tested, there was no difference in peak response between concentric squares, checkerboards, and OFF stimuli in either dendritic field (*Figure 6E and F*). As in the earlier data, ON stimuli produced a higher peak response in field C and a lower peak response in field A (p=0.015). The response at the SIZ showed no preference for ON, OFF, or mixed stimuli indicating similar spiking output from all four tested stimuli (*Figure 6G*).

## Energetic implications of ON/OFF dendritic mappings

To further explore the functional significance of segregating ON and OFF excitation between dendritic fields, we conducted a series of simulations on a biophysical model of the LGMD neuron. Previous LGMD models have reproduced many properties of the neuron, including responses to black looming stimuli (*Dewell and Gabbiani, 2018a*). The current model used the same pattern of synaptic inputs and model morphology (*Figure 7A*), with updated membrane parameters incorporating findings from subsequent investigations (*Dewell and Gabbiani, 2018b*; *Dewell and Gabbiani, 2019*).

The first simulation tested how inputs to fields A and C differ in their ability to initiate action potentials. Excitation to field A generated by a simulated black loom produced reliable firing (*Figure 7A and B*, blue). Moving the excitatory inputs from field A to C produced higher firing (*Figure 7B*, dark red). Reducing the number of field C excitatory synapses by 60% without changing the amount of inhibition still produced as much firing as the field A excitation simulation. This was due to the relative electrotonic proximity of field C to the SIZ; consequently, the depolarization elicited by synaptic inputs in field C attenuates less as it propagates toward the SIZ. This was confirmed by additional simulations in which the axial resistance was increased between field C and the primary neurite, after which as much excitation was needed in field C as in field A to elicit the same response.

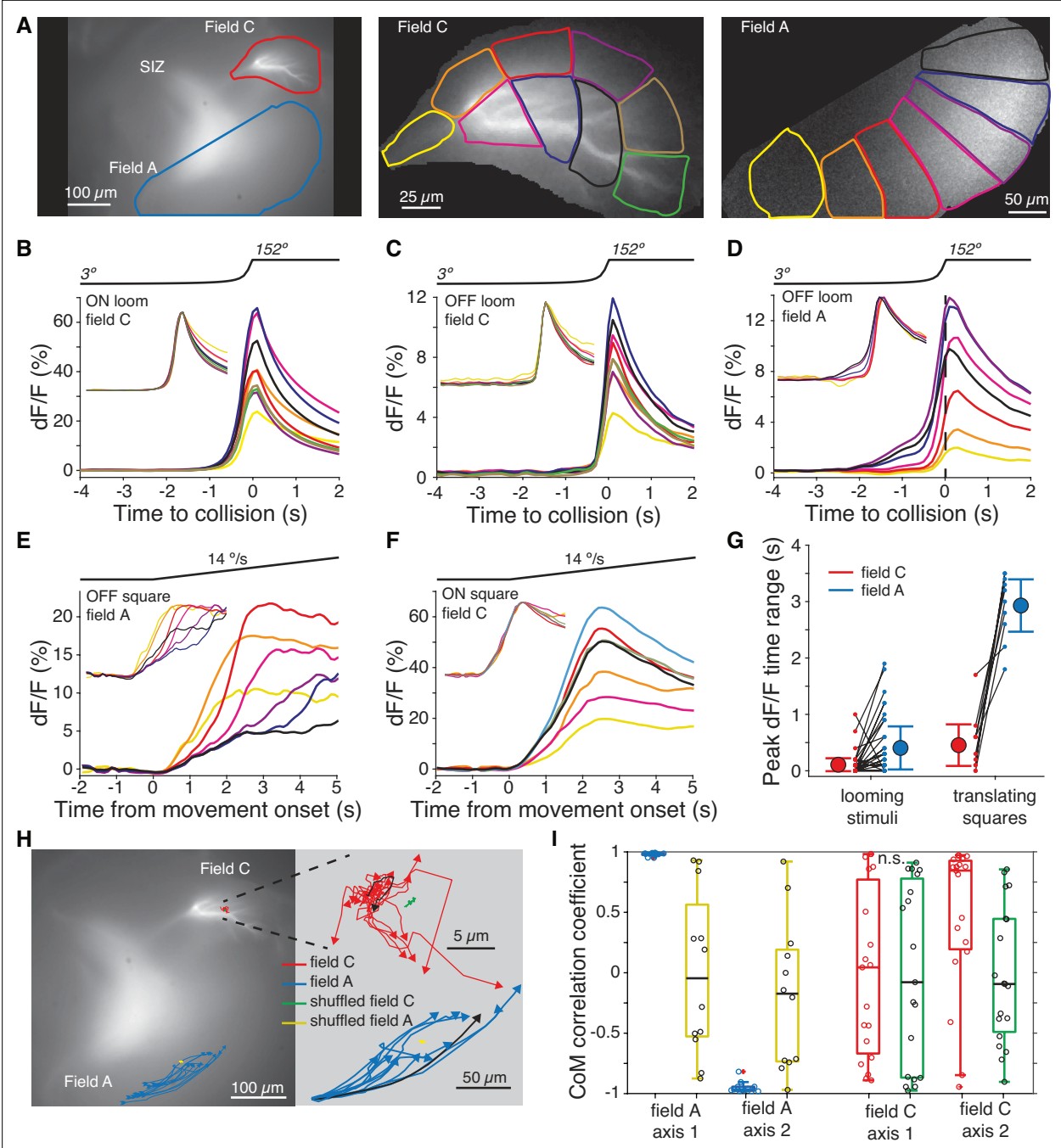

**Figure 4.** Within-field comparisons show lack of retinotopy in field C. (**A**) Left: example dF micrograph in response to an ON loom with dendritic fields marked. Middle: close-up of field C indicating the color-coded subregions used for dF/F calculations. Right: similar close-up of field A in response to a black loom. (**B**) Time course of mean dF/F in field C subregions shown in (**A**) in response to ON looms. The traces are rescaled to have the same peak value in the inset. (**C**) Black looms elicited similarly timed responses across field C subregions. (**D**) Black looms produced earlier and larger responses in field A subregions receiving inputs from the center of the loom. (**E**) Responses to black translating squares show sequential activation of field A subregions (inset, as in **B**). (**F**) ON translating squares produced a large synchronous dF/F signal across field C (inset, as in **B**). (**G**) The range in peak dF/F loom response times between subregions was higher in field A than C (p=3.2 • 10⁻⁴, t-test, N = 21). For translating squares, the range of dF/F peak times was larger for field A (p=0.002, t-test, N = 8). (**H**) Micrograph shown in (**A**) with superimposed trajectories of the dF/F center of mass in dendritic fields A and C for each trial of a translating bar (blue and red lines). For each dendritic field, the center of mass trajectories of randomized data are shown with yellow and green lines, respectively. At right, the same trajectories are shown zoomed in with each trajectory aligned to their initial position to illustrate the direction of spread. For each field, one example trajectory is shown in black. (**I**) The correlations between dF/F center of mass and bar position of each animal. Data from fields A and C are shown in blue and red, respectively, and randomized data for each is shown in yellow and green. Circles

*Figure 4 continued on next page*

*Figure 4 continued*

indicate data from individual trials. For (**H**) and (**I**), data is taken from 19 trials of five animals. See Methods, data analysis and statistics for a detailed description of box plots.

The online version of this article includes the following figure supplement(s) for figure 4:

**Figure supplement 1.** Micrographs of field C dendrites showing variability of branching patterns.

Calcium fluorescence showed that looms of all contrasts produced some excitation in both dendritic fields (*Figure 2*), so we next tested how the LGMD response changed with excitation split between the fields. For simulated looming stimuli, we varied the fraction of excitation distributed between the dendritic fields while removing 60% of the field C inputs since this achieved the same firing output regardless of dendritic field input assignment (see *Figure 7B*). The more evenly the inputs were split between the fields, the larger the resulting neural response was (*Figure 7C*). Thus, splitting of excitation between fields A and C maximizes its effectiveness at triggering LGMD spiking.

ON synaptic inputs show no retinotopy and are distributed throughout dendritic field C (*Figure 4*, *Video 6*). Yet, dendritic field C is sufficiently large electrotonically to allow for an approximate retinotopic mapping. Unlike field A, though, its shape does not match that of the eye (*Peron et al., 2007*), and is not consistent across animals (*Figure 4—figure supplement 1*). Consequently, there is no obvious visual input projection scheme leading to a retinotopic mapping in field C. Yet, retinotopy in field C would cause early excitation from a loom to be clustered and then spread across the dendrites from a central point. To test whether the lack of approximate retinotopy changed the LGMD's output, we simulated such clustered excitatory inputs that spread across field C (*Figure 7—figure supplement 1*). We found that looming excitation in field C with a similar spatial extent as that of field A reduced firing by 15% (*Figure 7D*). A narrower clustering of inputs further reduced responses. Notably, the observed reduction in loom responses translated into changes in escape behavior in previous reports (*Fotowat et al., 2011*; *Dewell and Gabbiani, 2018a*).

In earlier simulations, inputs to field A also produced larger responses when the retinotopic mapping was changed to a random one, provided the active membrane conductances responsible for coherence discrimination were removed (*Dewell and Gabbiani, 2018a*). Thus, in the absence of active membrane conductances, random synaptic localization of ON inputs in field C is more effective at triggering LGMD spiking than one approximating retinotopy. The biophysical explanation for this observation is that the more clustered synaptic inputs cause a larger reduction in synaptic current resulting from a decrease in driving force.

The LGMD remains sensitive to the spatial coherence of checkered stimuli despite a substantial fraction of excitation impinging on field C (*Figure 5*). To test whether the model reproduced this result, we simulated checkered looms of different coherence levels. In these simulations, excitation from OFF checkered regions impinged on field

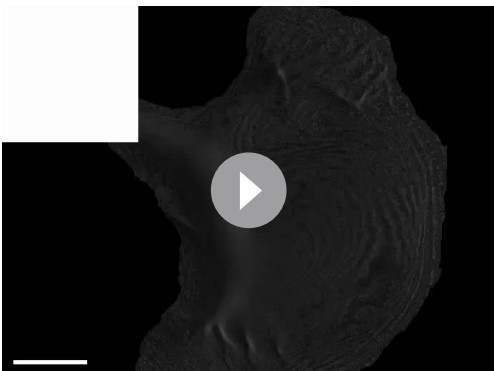

**Video 5.** Example of fluorescence dF/F within the lobula giant movement detector (LGMD) in response to a black moving dot stimulus (inset upper left). Scale bar: 100 μm. Note strong activation in eld A sweeping along a crescent starting from bottom center (out of focus). In addition, some uniform activation is also seen in eld C (top right, in focus). Matches data in *Figure 4E*.
https://elifesciences.org/articles/79772/figures#video5

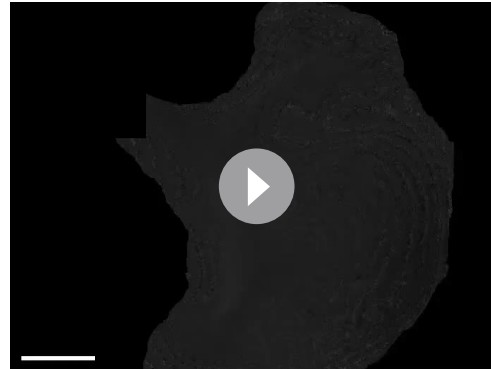

**Video 6.** The calculated fluorescence dF/F for a moving ON dot recorded from the animal with the least random field C mapping (see *Figure 4*). Scale bar: 100 μm.
https://elifesciences.org/articles/79772/figures#video6

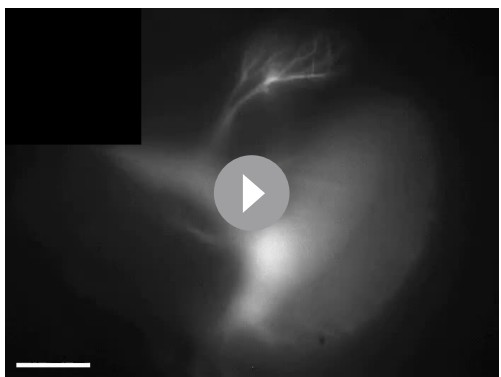

**Video 7.** An example of the raw fluorescence signal within the lobula giant movement detector (LGMD) during a white moving dot stimulus (inset upper left). As the object moves, the fluorescence increases in all of field C (top center, in focus). Data is from the same lobula giant movement detector (LGMD) as shown in *Videos 5 and 6*. Matches data in *Figure 4F*. https://elifesciences.org/articles/79772/figures#video7

A and ON regions excited field C. The model reproduced the decreased firing and increased bursting of spatially incoherent stimuli seen in experimental data (*Figure 7E*). The overall spatial coherence preference was qualitatively similar in model and experiment (*Figure 7F*). This confirms that for these mixed contrast stimuli, the retinotopic mapping and active filtering of field A produces spatial coherence selectivity even when 50% of the excitatory inputs are removed from field A.

Based on these simulation results, ON excitation impinging onto field C produces energetic savings compared to the previously hypothesized retinotopic mapping to field A. Considering the 60% reduction in excitatory inputs to field C due to their proximity to the SIZ, this represents a 30% total input reduction without losing sensitivity to spatial coherence for textured stimuli. As solid black looming stimuli partially excite field C (*Figure 2*), the simulations suggest that energetic savings of ~10% for black stimuli still generate the same spiking output. Stimuli with a mix of ON and OFF inputs would also benefit from higher savings than if all excitation impinged on field A.

## Discussion

### Dendritic segregation of ON/OFF excitation shapes looming detection and escape behavior

We investigated a looming-sensitive neuron that integrates ON and OFF inputs and responds to approaching objects with a characteristic firing profile invariant to object contrast (*Simmons and Rind, 1997*; *Gabbiani et al., 2001*). Although the LGMD neuron responds to either white or black approaching objects, whether grasshoppers escape from both was previously untested. We found that animals escaped from simulated white objects approaching a collision course (*Figure 1*) even if their spatial coherence was removed (*Figure 5*). The behavioral and physiological responses generally occurred slightly earlier for ON than OFF or checkered stimuli (*Figures 1 and 2*). Our results suggest that the ability to detect and respond to white objects but not to discriminate their spatial coherence is due to a previously unknown segregation of ON and OFF excitation and their distinct mapping onto separate dendritic fields. In previous experiments, relatively little attention was devoted to white looms, and there has been no test on the role of the LGMD in producing such escapes in contrast to that established for black stimuli (*Fotowat et al., 2011*). That the behavioral timing and coherence selectivity both match LGMD response changes suggests that the LGMD is initiating the jump escapes from both white and black looms.

Unlike the previously described OFF excitatory inputs that are retinotopically mapped with high precision on dendritic field A, the ON excitatory inputs impinge non-retinotopically onto field C (*Figure 4*). Both the retinotopic input mapping and the active conductances within field A are critical for OFF spatial coherence selectivity (*Dewell and Gabbiani, 2018a*; *Zhu et al., 2018*). Although it is not known whether field C is passive, it lacks the HCN channels present in field A that were found to be necessary for the animal's behavioral selectivity to the spatial coherence of black approaching stimuli (*Dewell and Gabbiani, 2018a*). Thus, our results demonstrate the computational and behavioral consequence of distinct dendritic synaptic input mappings and active conductance localization.

As real approaching predators likely exhibit nonuniform visual contrast, we also tested responses to stimuli with a mix of ON and OFF regions. Checkered stimuli produced responses very similar to solid black stimuli (*Figures 5 and 6*). The LGMD was responsive to checkered stimuli and discriminated their spatial coherence, demonstrating that only part of the excitation needs to impinge retinotopically

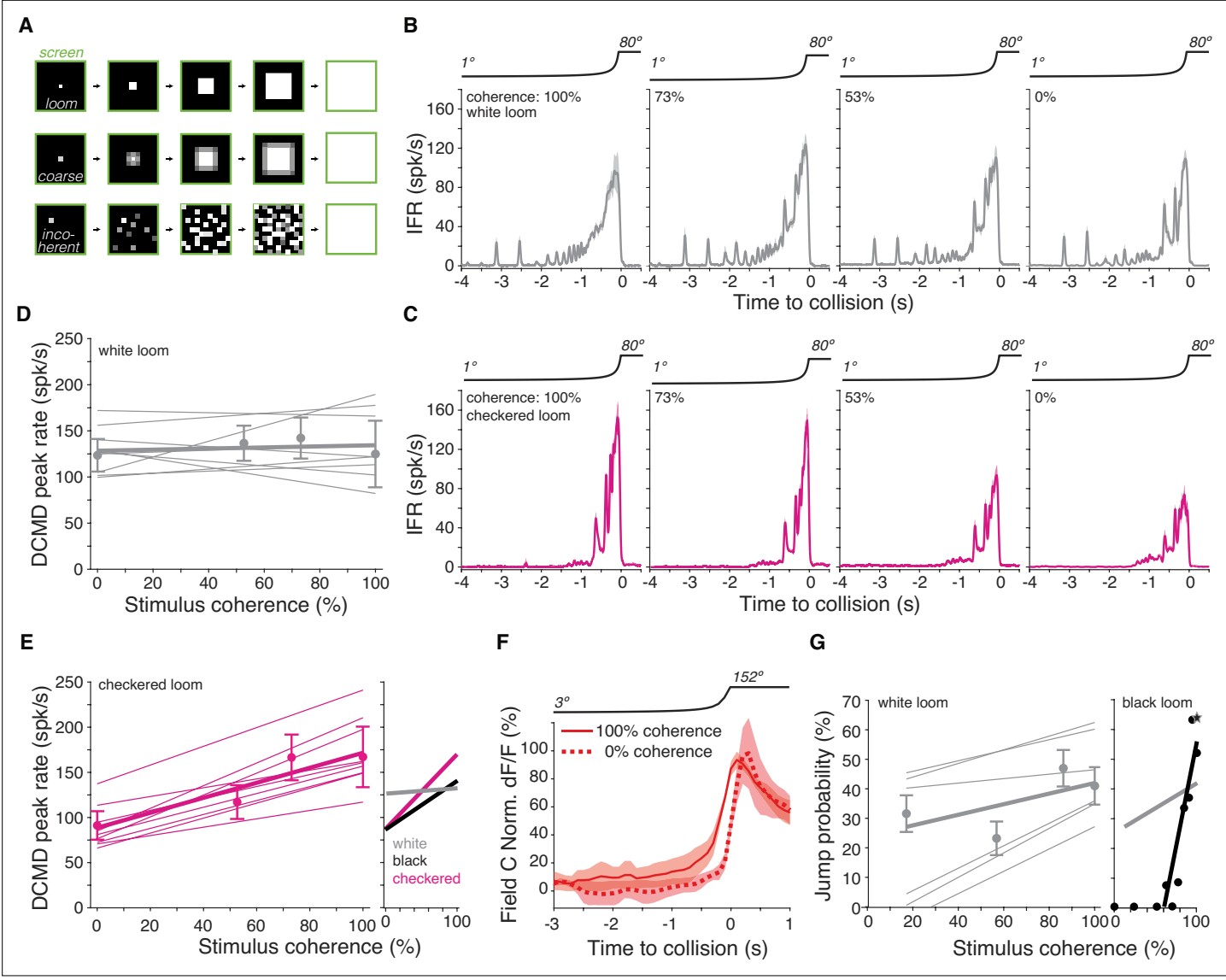

**Figure 5.** Lack of spatial coherence preference for white stimuli. (**A**) Top: schematic of looming stimulus. Middle: coarse looming stimulus. Bottom: stimulus with reduced spatial coherence. (**B, C**) Firing frequency of the lobula giant movement detector (LGMD) in response to white and checkerboard looming stimuli, respectively. Lines and shaded region are mean ± SEM, N = 9. As the spatial coherence was decreased, the response to checkered looms decreased (magenta). Responses to white looming stimuli did not decrease with reduction in spatial coherence (gray). (**D, E**) Linear fits of the peak firing rate to the spatial coherence of the stimulus. Thin lines are fits to individual animals, thick lines are fits to the population. Points and error bars are population median ± mad. Checkered stimuli had a mean $\rho$ of 0.91, while for white stimuli, mean $\rho$ was 0.15. Right inset: firing rate as a function of stimulus coherence for black looms (from ***Dewell and Gabbiani, 2018b***) compared to data for white and checkered looms at left and in (**D**). (**F**) Time course of field C normalized dF/F in response to 100 and 0% coherent white looming stimuli shows no change in peak value (N = 6). (**G**) Left: jump probability as a function of stimulus coherence for white looms (mean ± SD, N = 6). Thick gray line: linear fit; thin gray lines: fits on individual animals. Right inset: jump probability as a function of stimulus coherence for black looms (from ***Dewell and Gabbiani, 2018a***). Thick gray line is reproduced from left plot for comparison, but note these data are from different animals so the absolute jump probabilities cannot be directly compared.

The online version of this article includes the following figure supplement(s) for figure 5:

**Figure supplement 1.** Instantaneous firing rates for white and checkered stimuli from ***Figure 5*** compared to previously recorded responses to black stimuli.

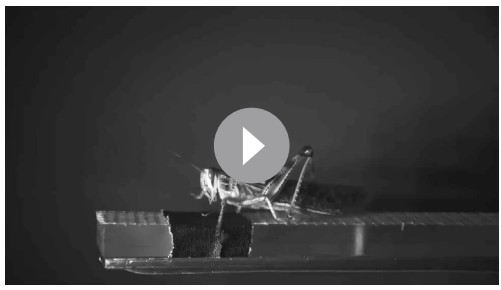

**Video 8.** Example response showing a grasshopper jump escape in response to a white 20% spatially coherent looming stimulus.
https://elifesciences.org/articles/79772/figures#video8

onto field A to evoke selective escape responses. The visual detection of approaching predators is believed to rely more on OFF than ON information (*Zhou et al., 2022*). Hence, the neural and behavioral selectivity for the spatial coherence of black and checkered stimuli but not white ones indicates that grasshoppers can discriminate the spatial patterns of real-world threats.

Although we cannot experimentally test the behavioral response of an animal lacking field C from the LGMD, another looming-sensitive neuron laying adjacent to the LGMD called the LGMD2 provides some insight. The LGMD2 has a dendritic field A of the same size and shape as that of the LGMD but lacks analogous fields B or C. It responds with a strong preference to dark objects and exhibits limited ability to detect white approaching stimuli (*Simmons and Rind, 1997*; *Rind and Leitinger, 2000*). This further suggests the necessity of the additional dendritic field and its ON excitatory inputs for the LGMD's detection of impending collisions independent of contrast polarity.

## Interpretation of calcium fluorescence

Both ON and OFF pathways excite the LGMD through calcium-permeable nAChRs (*Figure 3*; *Peron et al., 2009*). This allowed characterization of the functional segregation between fields with an intracellularly injected fluorescent calcium indicator ('Methods'). The exact relationship between the amount of synaptic excitation and dF/F remains unknown but does not affect qualitatively our interpretation of experimental results. The intracellular injection of the fluorescent dye typically yields an uneven fluorescence distribution within and between cells. This likely contributed to the variability observed across animals for ON vs. OFF selectivity in fields C and A, respectively (see, e.g., *Figure 2K*). Thus, the dF/F values indicate a general activity level of the dendrites, dependent on a

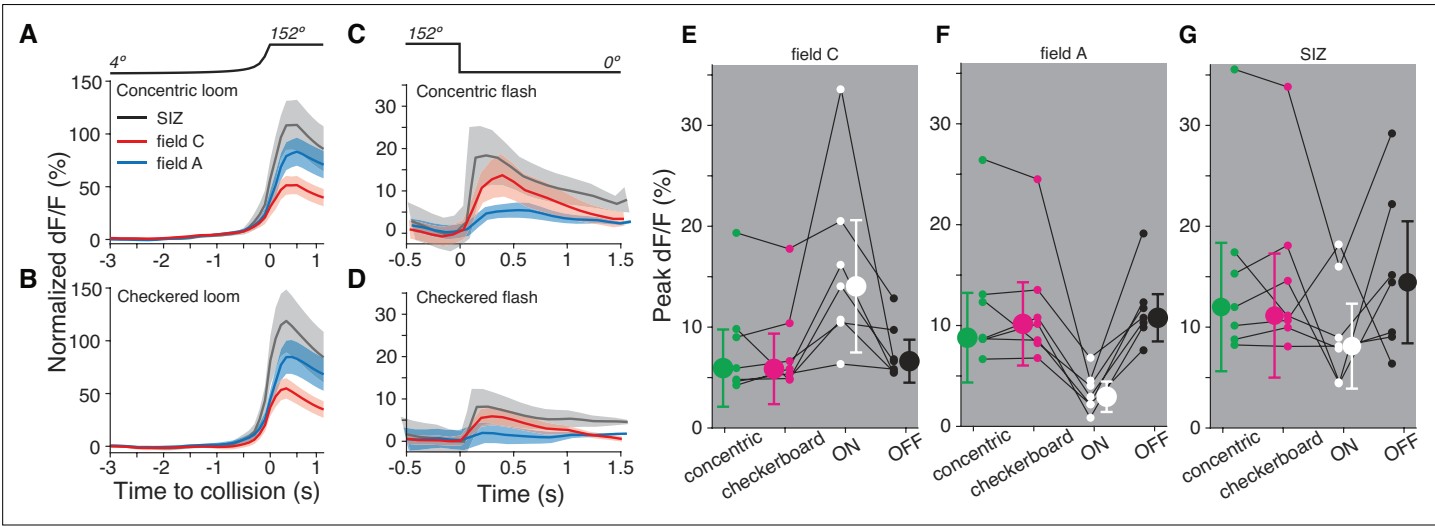

**Figure 6.** Both dendritic fields A and C respond similarly to looming stimuli of mixed luminance and to black looms. (**A**) Looming stimuli composed of five concentric ON and OFF squares produce slightly higher responses in field A (p=0.016, sign test [ST]). (**B**) Checkered looming stimuli similarly produced higher field A responses (p=0.016, ST). (**C, D**) Post-loom flash responses were higher in field C after concentric looming stimuli (p=0.026, ST) but not for post-checkered flashes (p=0.093, ST). (**E**) Field C peak dF/F for each stimulus. ON looms elicited the highest response in all animals (p=0.015, ST), and responses to the other three stimuli were not significantly different (p=0.82, Kruskal–Wallis test [KW]). (**F**) Field A peak dF/F for each stimulus. ON looms elicited the lowest response in all animals (p=0.015, ST), and responses to the other three stimuli were not different (p=0.89, KW). (**G**) All stimuli produced similar peak dF/F at the spike initiation zone (SIZ) (p=0.26, KW). In (**A–G**), N = 7 animals.

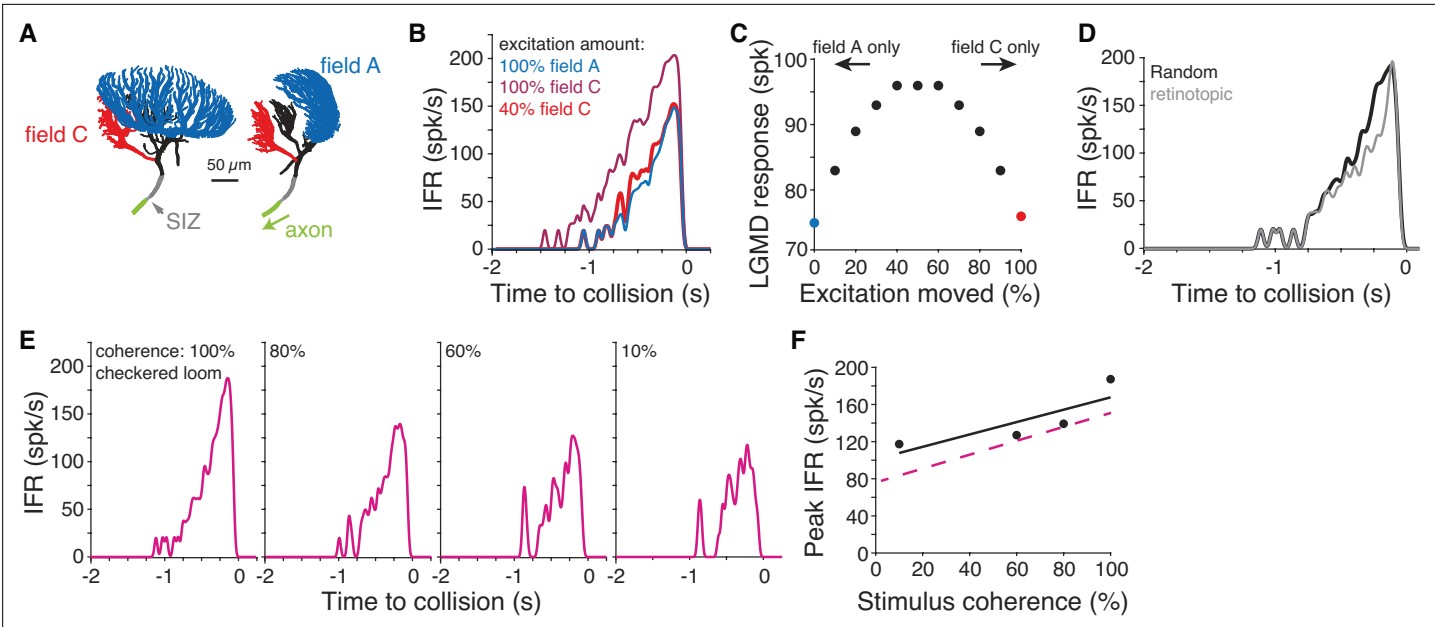

**Figure 7.** Simulations of a biophysical lobula giant movement detector (LGMD) model reveal energetic savings of segregated inputs. (**A**) Images of the model morphology showing the dendritic fields; the right image shows the model neuron rotated 90° from the left image. (**B**) Instantaneous firing rate (IFR) of the model's response to a black loom with excitation impinging on field A (blue line; same synapse locations as in *Dewell and Gabbiani, 2018b*). Moving all excitation to random field C locations (dark red) increased firing. Removal of 60% of excitatory synapses still produced a response as high as the field A inputs (red). (**C**) If excitatory synapses of simulated looms were distributed between the dendritic fields (with removal of 60% of synapses moved to field C), responses were highest with inputs split evenly between fields. Red and blue points are same as simulations shown in (**B**). (**D**) If all excitation impinged on field C with excitatory inputs clustered and spreading from a central point to approximate a retinotopic mapping, responses were reduced relative to inputs with random mapping. (**E**) Simulations reproduced firing patterns of response to checkered looming stimuli of different spatial coherences (*Figure 5C*). (**F**) Spatial coherence preference for checkered looms for the model (black) and experimental data (dashed magenta).

The online version of this article includes the following figure supplement(s) for figure 7:

**Figure supplement 1.** Illustration of retinotopic clustering used in simulations for *Figure 7D*.

combination of pre- and postsynaptic properties. For black looming stimuli, the dF/F is well correlated to the subthreshold membrane potential (*Zhu et al., 2018*).

## Visual processing of ON and OFF pathways

In mammals and insects, visual processing is similarly split between ON and OFF pathways. In both cases, photoreceptors respond to luminance changes of either polarity, but at the next stage increments and decrements are encoded by different neurons in the mammalian retina: the ON and OFF bipolar cell classes (*Euler et al., 2014*; *Clark and Demb, 2016*). In the lamina of the insect optic lobe, the large monopolar cells L1, L2, and L3 postsynaptic to photoreceptors respond differently to luminance and contrast changes of either polarity and distribute this information to ON and OFF neuron classes downstream that become increasingly selective (*Clark et al., 2011*; *Strother et al., 2014*; *Yang et al., 2016*). Further, large monopolar cells regulate contrast selectivity dynamically depending on background luminance levels (*Ketkar et al., 2020*; *Ketkar et al., 2022*). These observations stem from studies in vinegar flies under steady-state luminance conditions. However, during ON and OFF looming the mean luminance level is rapidly changing.

It remains unknown how this affects ON and OFF contrast selectivity in *Drosophila melanogaster* and *a fortiori* in *S. americana*. It is nonetheless safe to assume that contrast selectivity will be decreased under such conditions. Notably, we found a weaker contrast selectivity to looming stimuli in field C than in field A, mirroring the coarser spatial resolution expected in the excitatory projection to field C. Further, contrast selectivity was improved for both ON and OFF flashes that occurred after the loom when luminance was constant for 2 s, a condition that better approximates steady-state conditions. In summary, although contrast selectivity is weaker in field C than in field A and remains to be fully

characterized under the various stimulation conditions of this study, the most parsimonious explanation for responses of dendritic fields A and C to white and black stimuli is that they receive predominantly OFF and ON inputs, respectively, but that neither pathway is wholly selective for contrast polarity.

In both insects and mammals, the OFF pathway is more sensitive to fast movement than the ON pathway (*Leonhardt et al., 2016*; *Mazade et al., 2019*). In mammals OFF neurons are biased toward central vision (*Mulholland and Smith, 2021*; *Williams et al., 2021*). In insects, the columnar organization of the optic lobe likely produces an equal distribution of ON and OFF neurons. The central bias in mammals is related to the smaller receptive fields of OFF neurons (*Mazade et al., 2019*). In the grasshopper, the excitatory OFF inputs to field A of the LGMD have the same spatial resolution as the eye (~2°; *Jones and Gabbiani, 2010*). Feedforward inhibitory inputs to field C have lower resolution with receptive fields tenfold larger (*Wang et al., 2018a*; *Zhu et al., 2018*). The similarities of contrast polarity encoding between taxa likely result from visual processing being tuned to the statistics of natural scenes (*Clark et al., 2014*; *Clark and Demb, 2016*; *Chen et al., 2019*).

As this is the first report of ON excitation to field C, the inputs have not yet been characterized and little is known about the presynaptic circuitry. The known anatomy and current data suggest on the order of 15 times fewer presynaptic neurons with larger receptive fields than in field A (*Elphick et al., 1996*; *Zhu et al., 2018*). The DUB that projects to field C contains ~500 neurons that were previously believed to convey feedforward inhibitory input. Since inhibitory input is conveyed by a small set of neurons with larger receptive fields (*Wang et al., 2018a*), these DUB neurons are likely conveying the ON excitation described herein.

## Role of local integration of inhibitory and excitatory inputs in LGMD processing

Before this study, the LGMD was believed to integrate excitation and inhibition in separate dendritic subfields. That field C receives both excitatory and inhibitory inputs now raises questions about their local integration. Inhibition during a looming stimulus is thought to act mainly through shunting of the membrane resistance. Hence, the influence of OFF inhibition to field C will remain effective even if field C is somewhat excited during an OFF loom. Further, pharmacological block of field C inhibition showed that it mostly influences firing at the end of a loom when it causes the LGMD to stop firing (*Gabbiani et al., 2002*; *Gabbiani et al., 2005*). This termination of firing is thought to be important for triggering escape (*Fotowat et al., 2011*). Assuming that the selectivity of inhibition for ON/OFF polarity is similar to what we observed for excitation, the local integration of inhibition and excitation within field C might contribute to the earlier response peaks for ON looms. Future characterization of ON inhibition and the contrast selectivity of the inhibitory inputs will allow a more detailed examination of this topic.

## Energy costs of looming detection

The nervous system is an energetically expensive organ, estimated to consume over 20% of calories in humans. Reducing the cost of neural processing is thus critical for retinal and cortical circuits (*Vincent and Baddeley, 2003*; *Hasenstaub et al., 2010*). Neuronal energy expense increases with cell size, membrane conductance, and active conductances that pass $Na^+$ or $Ca^{2+}$ at rest, all of which suggests the LGMD is likely energetically expensive (*Hasenstaub et al., 2010*; *Niven, 2016*; *Zhukov et al., 2019*). Excitatory synaptic transmission contributes greatly to this energetic cost, accounting for about 2/3 of the brain's total ATP consumption (*Sibson et al., 1998*; *Attwell and Laughlin, 2001*; *Niven, 2016*; *Zhukov et al., 2019*). Most of that expense comes from the pumping of $Na^+$ and $Ca^{2+}$ ions out of neurons following excitatory synaptic events (*Attwell and Laughlin, 2001*; *Hasenstaub et al., 2010*). Thus, reducing the number of synaptic events needed to detect an approaching object could potentially generate large energetic savings.

We estimated the energetic cost of looming-evoked excitation from currents within the model. The model EPSPs carried an average charge of 0.34 pC (see 'Methods'). A simulated black loom had 80,000 EPSPs within field A, for a total charge of 27 nC or $1.7 \cdot 10^{11}$ ions (after multiplication by Avogadro's number and division by Faraday's constant). The $Na^+/K^+$ pump expels three $Na^+$ ions per ATP molecule, so if all excitatory current was from $Na^+$, $5.7 \cdot 10^{10}$ ATP molecules would be required to pump them out. Some excitatory current is passed by $Ca^{2+}$. Accounting for the efficiency of the

Na/Ca exchanger (one $Ca^{2+}$ ion per ATP molecule), our final estimate of energetic expense of a loom with all excitation impinging on field A is $6 \cdot 10^{10}$ ATP molecules. Modeling showed that moving all inputs from field A to field C allows a 60% reduction in excitation while generating the same output (*Figure 7*), corresponding to a savings $3.6 \cdot 10^{10}$ ATP molecules per loom if all excitation impinged on field C and half that for split excitation. The estimate of ATP expenses was confirmed using two alternative ways of calculation (see 'Methods').

These energetic savings are due to field C inputs impinging closer to the site of spike initiation. Further, this estimate assumes that the presynaptic circuitry for ON and OFF inputs expends equal amounts of energy and neglects differences in dendritic processing. As the ON inputs likely comprise fewer neurons, the energetic savings from moving some excitation to field C are likely higher than estimated here. The LGMD receives excitatory inputs from a full hemifield of view triggering a sustained rate of visually elicited and spontaneous excitation, so the savings would not be limited to synaptic excitation caused by approaching predators.

## Functional significance of synaptic mapping within dendrites

The additional energetic costs of OFF excitation to field A raise the question of why the LGMD would not have all inputs excite the more proximal field C. The proposed answer lies in the stark difference in selectivity of responses to ON and OFF stimuli. While the LGMD responds to white looms and the animals jump to escape them, the response is not selective to the spatial coherence of the stimulus (*Figure 5*). As escaping predation requires not just detecting threats, but discriminating them from nonthreatening stimuli, spatial coherence selectivity is likely critical for survival. As of yet, there has been no examination of whether mammalian looming-detection circuits are sensitive to stimulus spatial coherence or how they integrate inputs from ON and OFF pathways. Mice have a stronger behavioral response to black looms and looming-sensitive neurons in the superior colliculus show a preference for OFF inputs (*Yilmaz and Meister, 2013*; *Branco and Redgrave, 2020*).

Within the LGMD, the OFF pathway excites in precise retinotopic manner a distal dendritic field with large HCN and inactivating $K^+$ conductances that enable discrimination of spatiotemporal input patterns (*Zhu and Gabbiani, 2016*; *Dewell and Gabbiani, 2018a*). Conversely, the ON pathway results in non-retinotopic excitation of a proximal dendritic field lacking these active channels. Voltage-gated $Ca^{2+}$ channels, $Ca^{2+}$-dependent $K^+$ channels, and M-type $K^+$ channels that control bursting and spike-frequency adaptation are located close to the SIZ (*Peron and Gabbiani, 2009*; *Dewell and Gabbiani, 2018b*). The more distal position of field A makes integration within its dendrites further removed from the influence of these conductances and more dependent on the channels localized within it. The increased electrotonic distance from the SIZ also increases the resolving power of dendritic integration in response to synaptic input patterns since more excitatory inputs are required to generate action potentials.

The combination of reduced energy cost of field C inputs with increased discrimination by active processing in field A suggests a possible general principle for dendritic mapping whereby finer discriminations of the synaptic input pattern are implemented in distal, active dendrites while coarser discriminations are implemented by proximal dendrites. Additionally, having excitatory inputs spread between dendritic regions offers energetic savings of its own as concentrating inputs reduces the driving force of the activated receptor channels (*Figure 7C and D*). Examination of integration in neocortical or hippocampal pyramidal neurons that contain distinct dendritic regions and segregated inputs may provide a test for the generality of this mapping principle.

Within hippocampal pyramidal neurons, distal CA1 dendrites receive excitation from entorhinal cortex while proximal dendrites are excited by inputs from CA3. These dendritic regions also have distinct sets of active channels (*London and Häusser, 2005*; *Spruston, 2008*; *Lefebvre et al., 2015*). An alternative hypothesis to that put forth above suggests that proximal excitation is the primary driver of spiking while distal excitation is modulatory (*Behabadi et al., 2012*; *Hawkins and Ahmad, 2016*). The two ideas are not incompatible, though. Active processing in distal dendrites of cortical pyramidal neurons can produce fine discrimination of the spatiotemporal pattern of excitatory inputs even if dendritic compartmentalization and the location-dependence of synaptic integration are variable (*Poirazi and Papoutsi, 2020*).

Within cortex, feedback from higher cortical areas excites the distal apical dendrites of pyramidal neurons that also have increased active conductances that can produce fine discrimination of distal

inputs (*Schuman et al., 2021*). It has been suggested that the dendritic segregation of feedforward and feedback cortical inputs might underlie the comparison of sensory (bottom-up) inputs with top-down predictions (*Larkum, 2013*). In this context, our work suggests that distal feedback inputs might allow a finer discrimination of top-down than proximal bottom-up inputs. Within each neuron type, evolution presumably constrains the energetic costs of neural signaling required for dynamic, nonlinear computations. Systems in which we can simultaneously examine membrane properties, behavioral function, and dendritic mapping are crucial for discovering the underlying biophysical mechanisms.

## Methods

### Preparation

The experimental procedures used have been previously described (*Dewell and Gabbiani, 2018a*; *Zhu et al., 2018*). Experiments were conducted on adult *S. americana* grasshoppers 1–4 weeks after their final molt that were housed in a crowded laboratory colony. Preference was given to larger females ~3 weeks after final molt that were alert and responsive. Animals were selected for health and size without randomization. For calcium imaging, the head was opened to expose the brain and optic lobes. After removing the sheath protecting the right optic lobe, the calcium indicator Oregon Green Bapta (OGB-1) was injected by iontophoresis into the LGMD with sharp intracellular electrodes (*Zhu and Gabbiani, 2016*; *Zhu et al., 2018*). The amplitude and duration of current steps for OGB-1 ionto-phoresis were manually adjusted to produce a dim stain of the entire dendritic arbor. The variability in injection locations and baseline fluorescence levels produced a wide range of signal strengths across animals, with lower baseline fluorescence producing larger signal-to-noise ratios under visual stimulation.

### Pharmacology

For the pharmacology experiments described in *Figure 3*, ACh was delivered by iontophoresis to three animals and mecamylamine was pressure ejected to seven animals. In both application methods, drugs were applied to field C, but the pressure injection of mecamylamine caused spread to the dorsal half of field A while ACh iontophoresis remained localized close to the tip of the delivery pipette (*Figure 3—figure supplement 1*).

### Imaging

Calcium fluorescence was imaged with a CCD camera as previously described (*Zhu et al., 2018*). Images were captured through a ×16/0.8 numerical aperture (NA) water immersion objective lens (CFI75 LWD 16XW, Nikon Instruments) and saved at 5 Hz with a Rolera XR camera (Qimaging, Surrey, BC, Canada). The image resolution was 696 × 520 pixels and was saved as lossless 12-bit motion JPEG movies. The spatial resolution of the images was 0.9 µm per pixel. Experiments in which no change in OGB-1 fluorescence was seen in response to looming stimuli were excluded from analysis; this happened when too much OGB-1 was injected and the signal saturated or if not enough OGB-1 was injected to detect signals.

### Visual stimulation

For calcium imaging experiments, visual stimuli were generated with the Psychtoolbox (PTB-3) and MATLAB (MathWorks, Natick, MA) as done previously (*Zhu et al., 2018*). A digital light processing projector (DLP LightCrafter 4500, Texas Instruments, Dallas, TX) displayed stimuli on a screen (nonad-hesive stencil film, 0.1 mm thick) placed 20 mm from the right eye. The right eye was exposed, and the head and neck were submerged in saline. A mechanical brain holder was placed under the optic lobe to prevent movement perpendicular to the imaging plane. The DLP was programmed in pattern sequence mode to display 6-bit green-scale images with a refresh rate of 240 Hz and a 912 × 1140 pixel resolution.

For behavior and electrophysiology data shown in *Figures 1 and 5*, visual stimuli were generated by custom C software from a QNX4 computer and displayed on a CRT monitor with a 200 Hz refresh rate at a 640 × 480 pixel resolution (*Gabbiani et al., 1999*; *Dewell and Gabbiani, 2018a*). Stimuli for all experiments were displayed with 6-bit resolution luminance values. For behavioral experiments, looming stimuli had an $l/|v|$ of 40, 80, or 120 ms, and for calcium imaging stimuli with $l/|v|$ = 100 or

120 ms were used. The spatial coherence of looming stimuli was reduced by pixelating the stimuli into 2.5° regions and adding a spatial jitter to these coarse pixels (*Dewell and Gabbiani, 2018a*). For behavioral tests of coherence selectivity, the stimuli had an $l/|v|$ = 80 ms, and for the DCMD recordings an $l/|v|$=50ms.

## Behavior experiments

Jump experiments were conducted as previously (*Fotowat and Gabbiani, 2007*; *Dewell and Gabbiani, 2018a*). Adult grasshoppers of both sexes were used, and animals were presented stimuli in pseudo-random order, with at least 10 min between trials. Individual animals were presented up to 50 stimulus trials over up to 2 weeks of testing. Animals that did not jump to any stimuli were excluded from analysis. Trials in which the animal moved from the platform and did not see the stimulus were excluded from analysis. Videos were recorded with a high-speed digital video camera (GZL-CL-22C5M; Teledyne Flir), equipped with a variable zoom lens (M6Z 1212-3S; Computar, Cary, NC). Image frames were recorded at 200 frames per second with the acquisition of each frame synchronized to the vertical refresh of the visual stimulation display (Xtium-CL PX4; Teledyne Flir). Videos were made from the 12-bit images and saved in lossless motion JPEG format using custom MATLAB code.

## Data analysis and statistics

Sample sizes were not predetermined before experiments. All data analyses were carried out with custom MATLAB code (MathWorks). For calcium fluorescence, data was saved in either uncompressed AVI or lossless motion JPEG format using MATLAB's Image Acquisition toolbox. Most trials had no motion artifacts, but when present, translational motion artifacts were corrected by aligning dendritic locations across frames using the 'imregcorr' function of MATLAB's Image Processing toolbox. After any motion correction, videos were median filtered with a 3 × 3 pixel window to reduce noise. Fluorescence change (dF) was normalized to the baseline fluorescence (F), calculated as the average value in the 2 s before stimulus presentation to calculate dF/F. An example of the movement artifact can be seen in *Video 7*; in the raw data, there is translational movement due to breathing that was corrected before calculation of the dF/F. In some trials, the baseline fluorescence changed over time even in the absence of visual stimuli and a linear fit to the fluorescence time course before stimulus presentation was subtracted instead of a single value. Every trial was manually checked to confirm the change in fluorescence before any further data processing.

All regions of interest (ROIs) were drawn freehand, and the reported dF/F is the mean value of all pixels within the selected region. The period from the start of looming stimuli until 1 s after the end of expansion was considered the looming response. For post-loom flashes, the peak dF/F of the flash response was measured as the difference between the maximum dF/F within 1.5 s after the flash minus the minimum dF/F in the 1 s before the flash.

For testing whether there was a retinotopic mapping of inputs in field C, the center of mass of the dF/F within a dendritic field was calculated for each frame during the 4 s that the bar was moving using MATLAB's image processing toolbox (using the 'weighted centroid' property of 'regionprops'). The trajectory of the center of mass was then compared to the stimulus position at each corresponding time. To test the randomness of this mapping, each pixel of the dF/F within the dendritic field was randomly redistributed and the center of mass trajectory was recalculated.

The box plots in *Figures 1D and 4I* are displayed as follows. The central mark indicates the median, and the bottom and top edges of the box indicate the 25th and 75th percentiles, respectively. The whiskers extend to the most extreme data points not considered outliers (more than 1.5 times than interquartile range beyond the extent of the box), and the outliers are plotted individually using the '+' symbol.

For statistical tests between jump probabilities, we used Fisher's exact test to compare responses across stimulus speed and contrast. The timing of jump behavior was tested with the Wilcoxon rank-sum test. Paired comparisons of peak dF/F between dendritic fields or within dendritic fields for different stimulus polarities were made using a two-sided sign test. Comparisons across more than two stimuli (*Figure 6*) were made with the Kruskal–Wallis test, a nonparametric version of the classical analysis of variance (indicated by KW in text). All correlations were computed using Pearson's linear correlation coefficient.

## Neuronal modeling

We adapted a detailed biophysical LGMD model within the NEURON simulation environment that successfully reproduces many properties of the LGMD, including its 3D morphology, membrane conductances, and responses to a wide array of current injection protocols and visual stimuli (*Dewell and Gabbiani, 2018b*; *Dewell and Gabbiani, 2018a*; *Dewell and Gabbiani, 2019*). The previous model iterations are available from ModelDB (accession numbers 195666 and 256024). All previously simulated visual responses for this model were limited to OFF stimuli (dark objects on a light background). For the current investigations, we used this model to test the impact of ON/OFF excitatory segregation between fields A and C.

Briefly, the model morphology was generated from two-photon scans using the software suite Vaa3D (vaa3d.org), producing a resulting model with 2518 compartments, 1352 of which are in field A and 455 in field C. The membrane channels used in the model are the fast $Na^+$, delayed rectifier $K^+$, HCN, inactivating $K^+$ channels ($K_{D-like}$), KCNQ M-type $K^+$, low-threshold $Ca^{2+}$ ($Ca_T$), and $Ca^{2+}$-dependent $K^+$ ($K_{Ca}$), with kinetics and location of the channels constrained by experimental data (*Dewell and Gabbiani, 2018a*; *Dewell and Gabbiani, 2019*). All simulations were run with a time step of 0.02 ms using the standard NEURON integration algorithm.

The pattern of synaptic inputs for simulating responses to visual stimuli was generated as mentioned previously and is grounded in experimental data. The procedure for generating synaptic patterns consisted of aligning the stimulus to a virtual eye and calculating the time course of luminance within each facet's receptive field (*Dewell and Gabbiani, 2018a*). Signal transforms were applied to match the membrane potentials recorded from individual photoreceptors and laminar neurons (*Jones and Gabbiani, 2010*). For modeling spiking medullary neurons presynaptic to the LGMD, an additional thresholding was applied. The location of excitatory synapses impinging on field A was constrained by two-photon imaging data during single-facet simulation (*Zhu and Gabbiani, 2016*).

Each excitatory synapse was modeled as an alpha synapse with a 0.15 ms time constant, a maximal conductance of 14 nS, and a reversal potential of 10 mV. Each inhibitory synapse was modeled as an alpha synapse with a 2 ms time constant, a maximal conductance of 8 nS, and a reversal potential of −78 mV. There were randomly timed spontaneous synaptic inputs at rates of 500 Hz for excitatory inputs and 30 Hz for inhibitory inputs. This level of spontaneous activity reproduced the experimentally recorded noise level of the LGMD membrane potential (*Jones and Gabbiani, 2012a*). A simulated looming stimulus had 80,000 excitatory synaptic events and 7000 inhibitory events. Simulation time scales with the number of synapses, so the modeled inputs were implemented with fewer, higher conductance synapses than the actual excitatory inputs.

The model responses have been previously tuned to reproduce the timing and strength of experimentally recorded LGMD activity generated by single-facet stimulation, small and large flashes, moving bars, and looming stimuli (*Dewell and Gabbiani, 2018a*). The experimental data that was used for constraining the model's visual responses all used OFF stimuli. As equivalent data is not available for ON stimuli, we used the same synaptic inputs for ON and OFF stimuli except for the differences in dendritic locations and excitatory scaling as described in the section 'Results.' The excitatory inputs impinging on field C are believed to have larger receptive fields than those exciting field A due to fewer total neurons covering the visual hemifield. Future studies will be required to determine whether differences in receptive field properties influence the integration of ON and OFF inputs.

Previous simulations of black looming stimuli had excitation impinging on field A of the model and inhibition impinging on field C (*Jones and Gabbiani, 2012b*; *Dewell and Gabbiani, 2018a*). For simulations shown in *Figure 7* when excitation was moved from field A to field C, an equal percentage of inhibition was moved from field C to field B. For simulations comparing the relative strength of excitation in fields A and C shown in *Figure 7B*, we moved all excitation to pseudo-randomly selected field C compartments. After moving synapses to field C, their strength was changed by either reducing the conductance of each synaptic event or by removal of pseudo-random synaptic subsets. Both methods of reducing the field C excitation had the same effect, that is, a 60% reduction in conductance produced the same activity level as 60% event removal.

To test integration of split excitation between fields, we moved different pseudo-randomly selected subsets of field A excitatory inputs to field C (and a matching percentage of inhibitory inputs from field C to field B). The excitatory inputs moved to field C were then reduced by 60% (reducing conductance or removing events were both used with equal effect). These simulations were repeated with different

random subsets of synaptic events moved, resulting in similar responses (the range of spike count differences for the percentages tested were 0–2 spikes). The mean responses of these simulations are illustrated in *Figure 7C*.

To simulate responses to checkerboard stimuli (*Figure 7E and F*), the excitatory inputs from each facet viewing ON checkers (luminance increases) were mapped randomly to field C compartments and excitatory inputs from facets viewing OFF checkers were mapped retinotopically to field A. To simulate spatially scrambled checkerboard responses, 10 different stimuli were generated for each coherence level, and independent synaptic mappings were generated for each. The data in *Figure 7E and F* shows the average responses for each coherence level.

Electrotonic distance between field C and the SIZ was adjusted by increasing the axial resistance in the dendritic segment connecting field C to the main neurite that connects the axon to each dendritic field. The properties of the branches within field C were not altered.

To impose a retinotopic clustering on field C inputs (*Figure 7D*), we chose a compartment near the center of field C to serve as the 'center' of the loom response. The earliest looming inputs were set to impinge on this central point, and as the loom expanded the activated region of field C expanded based on a sigmoidal function with steepness of 80 μm. This resulted in a region of spread similar to that seen in experimental data for field A as illustrated by *Figure 7—figure supplement 1*.

### Alternative calculations of energy expenses

To confirm that the energy expenses presented in 'Discussion' are reasonable, we estimated them by two other ways – using published estimates of synaptic cost and extrapolating from current-clamp measurements.

First, the total energy cost of synaptic transmission is derived from

$$E_{tot} = E_{syn}N_{syn}N_{spk},$$

where $E_{syn}$ is the energy cost per synapse, $N_{syn}$ is the number of synapses, and $N_{spk}$ is the number of spikes of the presynaptic neurons per loom. Previous estimates for excitatory transmission have suggested $E_{syn}$ = 70,000 ATP molecules per synapse (*Attwell and Laughlin, 2001*). Field A of the LGMD has ~131,000 excitatory synapses (*Rind et al., 2016*) and our looming stimuli cover ~20% of the eye, suggesting $N_{syn}$ = 26,200. A value of $N_{spk}$ = 32 spikes produces a total energy cost ($E_{tot}$) of $6 \cdot 10^{10}$ ATP, equal to the model prediction. Past recordings of medulla neurons presynaptic to the LGMD found they fired ~40 spikes per loom, so this estimate would be in good agreement with our modeling results (*Wang et al., 2018b*).

The second check on the energetic cost estimate was to compute the total excitatory postsynaptic membrane current. Based on it, the estimated energy cost of a loom corresponds to an average excitatory current of 5 nA during the loom. Based on the rate of firing produced by injection of 5 nA into field A, this also matches experimental data (*Peron and Gabbiani, 2009*).

## Acknowledgements

We thank Drs. Alyse Thomas and Jake Reimer for manuscript feedback. We would like to thank the BCM Bioengineering Core for aid in experimental design. This work was supported by grants from the National Science Foundation (DMS-1120952 and DBI-2021795) and NEI Core Grant for Vision Research (EY-002520-37).

## Additional information

### Funding

| Funder | Grant reference number | Author |
| --- | --- | --- |
| National Science Foundation | IIS-1607518 | Fabrizio Gabbiani |
| National Science Foundation | DBI-2021795 | Fabrizio Gabbiani |

| Funder | Grant reference number | Author |
|---|---|---|
| National Eye Institute | EY-002520-37 | Fabrizio Gabbiani |

The funders had no role in study design, data collection and interpretation, or the decision to submit the work for publication.

## Author contributions

Richard Burkett Dewell, Conceptualization, Data curation, Software, Formal analysis, Supervision, Validation, Investigation, Visualization, Methodology, Writing - original draft, Writing – review and editing; Ying Zhu, Conceptualization, Formal analysis, Investigation, Visualization, Methodology, Writing – review and editing; Margaret Eisenbrandt, Formal analysis, Investigation; Richard Morse, Software, Formal analysis; Fabrizio Gabbiani, Supervision, Funding acquisition, Visualization, Writing - original draft, Project administration, Writing – review and editing

## Author ORCIDs

Richard Burkett Dewell http://orcid.org/0000-0003-2430-8184
Ying Zhu http://orcid.org/0000-0001-7188-1990
Margaret Eisenbrandt http://orcid.org/0000-0002-0190-1922
Fabrizio Gabbiani http://orcid.org/0000-0003-4966-3027

## Decision letter and Author response

Decision letter https://doi.org/10.7554/eLife.79772.sa1
Author response https://doi.org/10.7554/eLife.79772.sa2

# Additional files

## Supplementary files

• MDAR checklist

## Data availability

The data and code used to generate the final figures is available from Dryad (DOI: 10.5061/dryad.prr4xgxqp). The modeling code is available from ModelDB (Accession number: 267594).

The following datasets were generated:

| Author(s) | Year | Dataset title | Dataset URL | Database and Identifier |
|---|---|---|---|---|
| Dewell RB | 2022 | Contrast-polarity specific mapping improves efficiency of neuronal computation for collision detection | https://doi.org/10.5061/dryad.prr4xgxqp | Dryad Digital Repository, 10.5061/dryad.prr4xgxqp |
| Dewell RB, Gabbiani F | 2022 | LGMD - ON excitation to dendritic field C | https://senselab.med.yale.edu/modeldb/ShowModel?model=267594 | ModelDB, 267594 |

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
