## [Editor Report]

This valuable article will be of interest to neuroscientists who study visual processing or are interested in dendritic integration. The authors used calcium imaging, pharmacology, and electrophysiology to investigate how a large, loom-sensitive neuron in grasshoppers integrates visual input to respond to both light and dark looming objects. These experiments convincingly support the finding that the integration is done by two distinct arbors of the neuronal dendritic tree, one of which loses retinotopic information. The authors suggest energetic advantages of this dendritic arrangement.

---

## [Decision Letter]

**Decision letter after peer review:**

Thank you for submitting your article "Contrast-polarity specific mapping optimizes neuronal computation for collision detection" for consideration by *eLife*. Your article has been reviewed by 3 peer reviewers, one of whom is a member of our Board of Reviewing Editors, and the evaluation has been overseen by Ronald Calabrese as the Senior Editor. The following individual involved in review of your submission has agreed to reveal their identity: Tiago Branco (Reviewer #3).

Essential revisions:

The reviewers agreed that this manuscript would be suitable for publication in *eLife* if the authors address the points below.

1) Retinotopic mapping in field C: It remains unclear how the columnar organization of the lobula could relate to the proposed lack of retinotopy in field C. Similarly, the ROI based retinotopic analysis should be related to this anatomy. Both of these issues should be addressed.

2) Modeling: A detailed modeling section is required in the methods, including how excitation and inhibition are included.

3) Energetics: These claims need to be better supported by direct calculations and clear comparisons.

4) Optimizing claims: These must be better supported if they are to be included in the manuscript and title. As it stands, the claim is not backed up with concrete evidence that this anatomical organization optimizes the computation.

5) Inhibition: The contribution of inhibition should be considered when interpreting data and expanded on in the discussion.

6) Claims: Several smaller claims should be softened, as indicated in the detailed comments.

*Reviewer #1 (Recommendations for the authors):*

There were a few items that seemed incomplete in the manuscript:

1) There were a few places where claims were perhaps a little too strong:

a. Lines 152-161: Ach experiments. In principle, all these results are also consistent with an upstream neuron being excited by ach and using a different neurotransmitter to get the excitation to the LGMD. One solution is to simply adjust text. Or could one show that puffing a different excitatory neurotransmitter doesn't cause this?

b. Figure 4I: The authors say flatly that there is no retinotopic mapping in field C (line 191) and that the input locations are not different from random, but 4I shows that's not true along 1 axis in field C, even if the effect is small. (Or at least, n.s. is only marked for one of two axes in this panel.)

c. The authors assume that LGMD is mediating behavioral responses to ON looms. Is that proven or could it be here? If not, a caveat might be warranted.

2) There were two particular results that I thought deserved more explanation:

a. Field C really responds reasonably to the OFF loom. It's a little hard to reconcile this with treating it as the ON loom responsive field. Relatedly, on line 356: Incomplete rectification doesn't seem like it could explain the lack of selectivity in these fields. Incompletely rectified ON neurons would not also depolarize in response to contrast decrements, for instance. It seems like there would need to be a different parsimonious explanation.

b. Why does mecamylamine application to field C reduce the field A ON response?

3) The modeling and energetics arguments need substantially more detail to be convincing.

a. The authors should provide some details of the model in the methods. It would be helpful to have the basics of the model here; right now, there is nothing. Particularly important are the input transformations used to create the ON and OFF signals for the two fields – are these tonic or transient inputs? It could make a large difference to the energetics of the system.

b. Figure 7D: I would call 'clustered' 'retinotopic' if I'm understanding this correctly.

c. The noise in several modeling traces seems to be the same, which suggests this hasn't been averaged over noise (if there's noise in the model) or averaged over looms at different spatial locations. In particular, the noise in traces appears the same in the red and blue traces in 7B. But also in the black and gray traces in 7D. Is the gray really less than the black? The noise appears the same in these two traces (around t = -1), which suggests that this simulation is not averaging over noise instantiations or stimuli as it probably should. If this figure is the argument for why evolution dropped retinotopy to Field C, it's interesting, but it should be made stronger, perhaps with error bars or confidence intervals.

d. The energetics arguments should be spelled out more clearly:

i. When the authors write that there would be X savings for the ON Field C arrangement, they need to be clear about what the comparison is to. What is the alternative arrangement they're considering? ON and OFF retinotopic inputs to Field A? No ON inputs at all? I did not find this clear.

ii. There's a mix of detailed biophysical modeling and back-of-the-envelope style reasoning to come up with various numbers for the savings (in percents and in ATP). Doesn't the biophysical model allow one to compute the energy requirements directly and precisely?

*Reviewer #2 (Recommendations for the authors):*

1. Polarity does not reduce jump probability (Line 91):

a. The data in figure 1B show a trend towards a reduction in jump probability to white vs dark looms. Lower response rates to white vs dark stimuli have been documented across animals (Yilmaz and Meister 2013, Holmqvist and Srinivasan 1991, etc.), and figure 5G, at 100% coherence, seems to support this, with a >60% response rate for black and ~40% response rate for white looming stimuli. How do the Figure 1B data appear when using paired, individual animal probabilities as the data points, instead of 263 trials from 7 grasshoppers? Are the trials evenly distributed across the 7 grasshoppers?

2. The contribution of off inhibition for dendrite field C:

a. Dendrite field C, in addition to receiving on excitation, has been documented to receive OFF inhibition. Could a release of OFF inhibition (inhibitory rebound, for example) also be contributing to the response observed in dendrite C during white loom presentations? The authors should comment on this in the discussion. Additionally, why would off inhibition and on excitation be converging onto this particular dendrite field?

3. Lack of retinotopy for excitatory on inputs to dendrite field C:

a. How are the branches of dendrite field C arranged with respect to the columnar structure of the lobula? Dendrite field A spans across the lobula columns so adjacent areas in space are mapped to adjacent areas on the dendrite. In comparison, does the anatomy of dendrite field C provide insight into why an absence of retinotopic arrangement is observed? Are there substructures in the locust lobula where the different dendrite fields reside (Rosner 2017)? Can the authors expand on this, including what has been observed in past anatomical investigations?

b. Additionally, in analyzing the retinotopy with calcium imaging, can the ROIs for dendrite field C (Figure 4A) be drawn to match the columnar structure of the lobula as in field A, and/or drawn to follow the individual branches instead of combining multiple primary and secondary branches that may result in averaging the signal across multiple regions? The authors note they employed different strategies for drawing the ROI and witnessed similar results, but it would be helpful to include the ROI data that follow the analysis for dendrite field A. Finally, does the smaller size and location of dendrite field C suggest it samples from a much smaller (or different) receptive field than dendrite A? This relates back to (3) where it would be helpful to expand the introduction of the anatomy for the three dendrite fields, and expand on how it aligns or does not align with the ca^2+^ imaging results in the Discussion section.

4. Modeling synapse numbers as a proxy for energetic cost:

a. There is not a modeling section in the methods. The authors need to recap the methods for the published model and then describe in detail what changes/additions were made for the current manuscripts.

b. It is indeed interesting that the same firing rate can be achieved after removing 60% of dendrite field C's input, and that this may reduce energetic costs for the LGMD. However, are the starting number of synapses anatomically relevant?

---

## [Author Response]

Essential revisions:The reviewers agreed that this manuscript would be suitable for publication in eLife if the authors address the points below.1) Retinotopic mapping in field C: It remains unclear how the columnar organization of the lobula could relate to the proposed lack of retinotopy in field C. Similarly, the ROI based retinotopic analysis should be related to this anatomy. Both of these issues should be addressed.

Field C of the LGMD arborizes in a different sub-compartment of the lobula than field A, called the dorsal lobula, which is not thought to be retinotopically organized. Specifically, the previously described field C inputs originate from a bundle of fibers called the dorsal uncrossed bundle (DUB) which projects to the dorsal lobula (O’Shea and Williams, 1974). In contrast to the second optic chiasm between the medulla and lobula, which is a crossed bundle and provides a columnar and retinotopic projection, the DUB is an uncrossed bundle, providing a non-columnar and nonretinotopic projection (Gouranton, 1964; Strausfeld and Nässel, 1981, Figure 67; Elphick et al., 1996, p. 2396). The DUB projection was first described by Gouranton (1964; Figure 1, where it is called fnc2). Further anatomical work on Golgi stains confirmed that LGMD inputs to field C from the DUB are non-retinotopic (Strausfeld and Nässel, 1981, p. 119).

Additionally, recent work in our lab has shown that only a small number of LGMD inputs to field C originating from the DUB are inhibitory (on the order of 10 neurons; Wang et al. 2018). Thus, the novel ON excitation to field C described here most likely represent the majority of the ~500 non-retinotopic DUB inputs to field C, with receptive fields estimated to be around 8x12º (Rowell et al., 1977; Elphick et al., 1996). In agreement with the anatomical assumption of a few hundred non-retinotopic inputs with wider receptive fields than the columnar projections impinging on field A (~2x2º), modeling the inputs in this way was sufficient to accurately reproduce the data.

To make this clear we have added one paragraph in the introduction summarizing these results (lines 68-79).

Please see revised manuscript for all cited references.

2) Modeling: A detailed modeling section is required in the methods, including how excitation and inhibition are included.

A new section describing modeling detail has been added to the methods (lines 665-740).

3) Energetics: These claims need to be better supported by direct calculations and clear comparisons.

The corresponding calculations and details were added (lines 488-500 and 742-759).

4) Optimizing claims: These must be better supported if they are to be included in the manuscript and title. As it stands, the claim is not backed up with concrete evidence that this anatomical organization optimizes the computation.

We agree that optimality is a strong claim to make. We have modified the title and text accordingly to emphasize an ‘improvement’ in neural computation rather than its ‘optimization’.

5) Inhibition: The contribution of inhibition should be considered when interpreting data and expanded on in the discussion.

The manuscript does not propose any changes to the described inhibitory inputs, and the modeling section uses the same inhibition as our previous publications. Thus, inhibition is mostly out of the scope of the paper, but a short paragraph discussing its potential impact was added to the discussion (lines 461-473). Better characterizing inhibition and its interaction with excitation is high on our list of future experiments, now that we know that they both impinge on field C. There is no detailed description of the field B ON inhibitory inputs, so the current assumption that the ON and OFF feed-forward inhibition are of the same timing and strength is the simplest assumption.

6) Claims: Several smaller claims should be softened, as indicated in the detailed comments.

We have addressed these issues as detailed below.

Reviewer #1 (Recommendations for the authors):There were a few items that seemed incomplete in the manuscript:1) There were a few places where claims were perhaps a little too strong:a. Lines 152-161: Ach experiments. In principle, all these results are also consistent with an upstream neuron being excited by ach and using a different neurotransmitter to get the excitation to the LGMD. One solution is to simply adjust text. Or could one show that puffing a different excitatory neurotransmitter doesn't cause this?

The proposed scenario is theoretically possible, but the data would be very difficult to explain through such an indirect mechanism. Acetylcholine was delivered through iontophoresis directly at field C dendrites. The observed fluorescence change in field C is local, within ~25 µm of the electrode used for iontophoresis. A new supplemental figure showing this has been added (Figure 3S1), and the related text in the Results has been reworded to improve clarity (lines 184-189). If the field C activity was produced by activation of intermediate neurons, their synaptic inputs to field C would have to be perfectly aligned at the same location as the electrode across experiments, which is implausible.

Another hypothesis that would be consistent with the data is that the presynaptic excitatory neurons projecting to the LGMD field C have nAChRs on their presynaptic axon terminals and that these receptors must be activated to produce synaptic excitation of the LGMD. Although muscarinic acetylcholine receptors have been reported at presynaptic terminals, nicotinic receptors have never been reported there to the best of our knowledge.

Additionally, to the best of our knowledge, acetylcholine is the main excitatory neurotransmitter in the insect brain and we would not know which alterative excitatory neurotransmitter could be selected for puffing. Note that glutamate is an inhibitory neurotransmitter in the insect nervous system.

b. Figure 4I: The authors say flatly that there is no retinotopic mapping in field C (line 191) and that the input locations are not different from random, but 4I shows that's not true along 1 axis in field C, even if the effect is small. (Or at least, n.s. is only marked for one of two axes in this panel.)

We are confident that the mapping is not retinotopic, which matches the expectation from anatomy. This may be best seen by looking at videos 5-7 of moving dot stimuli and comparing the activation of field C and field A. We have changed the text to reduce claims of randomness, and focus on the point that there is no retinotopy. The corresponding Results section (lines 226-237) hopefully better clarifies our claims.

c. The authors assume that LGMD is mediating behavioral responses to ON looms. Is that proven or could it be here? If not, a caveat might be warranted.

Thank you. Earlier work has shown a causal link between LGMD activity and OFF looms, but not for ON looms. The text was changed to explain that the behavioral role of the LGMD for ON loom responses has not been directly confirmed (lines 371-376).

2) There were two particular results that I thought deserved more explanation:a. Field C really responds reasonably to the OFF loom. It's a little hard to reconcile this with treating it as the ON loom responsive field. Relatedly, on line 356: Incomplete rectification doesn't seem like it could explain the lack of selectivity in these fields. Incompletely rectified ON neurons would not also depolarize in response to contrast decrements, for instance. It seems like there would need to be a different parsimonious explanation.

There is no doubt in our mind from the experimental data that field C has a strong preference for ON stimuli and vice-versa for field A based on the summary data depicted in Figure 2K and L. Please see also videos 2-4. We agree that the wording ‘incomplete rectification’ was not optimal and it has been changed. We also agree that responses to field C are less selective than those to field A. We have expanded the discussion to explain that the exact selectivity likely depends on stimulus type and requires further study (lines 430-440). Another likely source of differences between the two fields may originate in technical aspects of dye injection as now better explained (lines 405415). We have toned down the claims as well.

b. Why does mecamylamine application to field C reduce the field A ON response?

In these experiments, mecamylamine was applied by pressure injection resulting in some spread to field A. We have added a section in the Methods explaining the details of drug application that addresses this point (lines 576-581).

3) The modeling and energetics arguments need substantially more detail to be convincing.a. The authors should provide some details of the model in the methods. It would be helpful to have the basics of the model here; right now, there is nothing. Particularly important are the input transformations used to create the ON and OFF signals for the two fields – are these tonic or transient inputs? It could make a large difference to the energetics of the system.

A detailed modeling section was added in the Methods (lines 665-740). The inputs are transient. The energetics details have been added as well (lines 488-50 and 742-759).

b. Figure 7D: I would call 'clustered' 'retinotopic' if I'm understanding this correctly.

We changed the label in Figure 7D.

c. The noise in several modeling traces seems to be the same, which suggests this hasn't been averaged over noise (if there's noise in the model) or averaged over looms at different spatial locations. In particular, the noise in traces appears the same in the red and blue traces in 7B. But also in the black and gray traces in 7D. Is the gray really less than the black? The noise appears the same in these two traces (around t = -1), which suggests that this simulation is not averaging over noise instantiations or stimuli as it probably should. If this figure is the argument for why evolution dropped retinotopy to Field C, it's interesting, but it should be made stronger, perhaps with error bars or confidence intervals.

There is some noise in the model (see added description in the Methods, lines 693-695) but we did not explicitly test the role of noise in these simulations as it is outside the scope of the present manuscript (but see Jones and Gabbiani J Neurophysiol 107:1067-79, 2012, for a characterization of noise in field A). A subset of the synapses in the model have random timing, but this does not much change the IFR profile. What appears as noise in the red and blue traces of 7B is actually consistent spike timing across trials (these are Gaussian smoothed firing rates). The consistency in spike timing is similar to that of experimental data, though. For comparison, see the experimental IFR and SD across animals in Figure 5B and C. The early spikes are reliably initiated at a set point of the loom and the model firing reflects this consistency.

d. The energetics arguments should be spelled out more clearly:

The energetics is explained in more detail (lines 488-507 and 742-759).

i. When the authors write that there would be X savings for the ON Field C arrangement, they need to be clear about what the comparison is to. What is the alternative arrangement they're considering? ON and OFF retinotopic inputs to Field A? No ON inputs at all? I did not find this clear.

The comparison is between the currently described ON inputs going to field C vs. the previously hypothesized retinotopic field A excitation. We clarified this in the text (lines 488-507).

ii. There's a mix of detailed biophysical modeling and back-of-the-envelope style reasoning to come up with various numbers for the savings (in percents and in ATP). Doesn't the biophysical model allow one to compute the energy requirements directly and precisely?

We computed the energetic costs from the model and show that the modeling agrees with experimental data. An exact energy budget would require data not currently available, so we initially reported only approximate values. More detailed descriptions of how energy was calculated and checked are now added (lines 488-501 and 742-759).

Reviewer #2 (Recommendations for the authors):1. Polarity does not reduce jump probability (Line 91):a. The data in figure 1B show a trend towards a reduction in jump probability to white vs dark looms. Lower response rates to white vs dark stimuli have been documented across animals (Yilmaz and Meister 2013, Holmqvist and Srinivasan 1991, etc.), and figure 5G, at 100% coherence, seems to support this, with a >60% response rate for black and ~40% response rate for white looming stimuli. How do the Figure 1B data appear when using paired, individual animal probabilities as the data points, instead of 263 trials from 7 grasshoppers? Are the trials evenly distributed across the 7 grasshoppers?

We added a new figure panel (1C) showing data for each animal, and the paired statistics showed a significant preference for black looms. With more data, the pooled population differences would likely be significant, but the polarity difference is much smaller in grasshoppers than in mice.

The absolute jump percentage for white and black stimuli in 5G can't be directly compared as the white and black responses are taken from different animals in separate experiments. The figure legend was updated to note this point.

2. The contribution of off inhibition for dendrite field C:a. Dendrite field C, in addition to receiving on excitation, has been documented to receive OFF inhibition. Could a release of OFF inhibition (inhibitory rebound, for example) also be contributing to the response observed in dendrite C during white loom presentations? The authors should comment on this in the discussion. Additionally, why would off inhibition and on excitation be converging onto this particular dendrite field?

This is an interesting question that will require additional experiments to be fully addressed. Based on our current knowledge, excitation precedes inhibition and inhibition does not stop until after collision (well after the end of LGMD firing). So, an inhibitory rebound is unlikely to contribute to the depolarization based on its expected timing.

A short paragraph on local integration of excitation and inhibition was added to the discussion (lines 461-473).

3. Lack of retinotopy for excitatory on inputs to dendrite field C:a. How are the branches of dendrite field C arranged with respect to the columnar structure of the lobula? Dendrite field A spans across the lobula columns so adjacent areas in space are mapped to adjacent areas on the dendrite. In comparison, does the anatomy of dendrite field C provide insight into why an absence of retinotopic arrangement is observed? Are there substructures in the locust lobula where the different dendrite fields reside (Rosner 2017)? Can the authors expand on this, including what has been observed in past anatomical investigations?

Anatomy does provide insight on the absence of retinotopic arrangement. Field C is located in different sub-compartment of the lobula called the dorsal lobula. This information has been added in the introduction (lines 68-79) and an expanded discussion of field C anatomy and DUB projections has been added in the same paragraph. Additionally, we added a new supplemental figure (Figure 4 – S1) showing that field C lacks the consistent branching pattern of field A.

b. Additionally, in analyzing the retinotopy with calcium imaging, can the ROIs for dendrite field C (Figure 4A) be drawn to match the columnar structure of the lobula as in field A, and/or drawn to follow the individual branches instead of combining multiple primary and secondary branches that may result in averaging the signal across multiple regions? The authors note they employed different strategies for drawing the ROI and witnessed similar results, but it would be helpful to include the ROI data that follow the analysis for dendrite field A. Finally, does the smaller size and location of dendrite field C suggest it samples from a much smaller (or different) receptive field than dendrite A? This relates back to (3) where it would be helpful to expand the introduction of the anatomy for the three dendrite fields, and expand on how it aligns or does not align with the ca^2+^ imaging results in the Discussion section.

We have expanded the introduction to explain that field C arborizes in a different part of the lobula (lines 68-79). Given the variable structure of field C (new Figure 4-S1), it is not possible to use the same strategy to segment it as used in field A. There is no indication from our data that field C samples a smaller visual region than field A, in particular small moving dots at any visual location tested elicited responses in both field A or C.

4. Modeling synapse numbers as a proxy for energetic cost:a. There is not a modeling section in the methods. The authors need to recap the methods for the published model and then describe in detail what changes/additions were made for the current manuscripts.

A detailed modeling section has been added to the methods (lines 665-740).

b. It is indeed interesting that the same firing rate can be achieved after removing 60% of dendrite field C's input, and that this may reduce energetic costs for the LGMD. However, are the starting number of synapses anatomically relevant?

The exact number of synapses used in the model are likely lower than the real neuron (lines 690699), but the spatial distribution of synapses and total synaptic current has been modeled based on extensive anatomical and physiological data as is now explained in the methods. The estimates from these calculations are also in good agreement with estimates from published data (lines 742-759).